# A phenome-wide association and Mendelian Randomisation study of polygenic risk for depression in UK Biobank

Xueyi Shen [1], David M. Howard [1,2], Mark J. Adams [1], W. David Hill[3,4], Toni-Kim Clarke[1], Major Depressive Disorder Working Group of the Psychiatric Genomics Consortium*, Ian J. Deary[3,4], Heather C. Whalley [1,153] & Andrew M. McIntosh [1,3,4,153 ✉]

Depression is a leading cause of worldwide disability but there remains considerable uncertainty regarding its neural and behavioural associations. Here, using non-overlapping Psychiatric Genomics Consortium (PGC) datasets as a reference, we estimate polygenic risk scores for depression (depression-PRS) in a discovery ($N = 10{,}674$) and replication ($N = 11{,}214$) imaging sample from UK Biobank. We report 77 traits that are significantly associated with depression-PRS, in both discovery and replication analyses. Mendelian Randomisation analysis supports a potential causal effect of liability to depression on brain white matter microstructure ($\beta$: 0.125 to 0.868, $p_{FDR} < 0.043$). Several behavioural traits are also associated with depression-PRS ($\beta$: 0.014 to 0.180, $p_{FDR}$: 0.049 to $1.28 \times 10^{-14}$) and we find a significant and positive interaction between depression-PRS and adverse environmental exposures on mental health outcomes. This study reveals replicable associations between depression-PRS and white matter microstructure. Our results indicate that white matter microstructure differences may be a causal consequence of liability to depression.

[1] Division of Psychiatry, University of Edinburgh, Edinburgh, UK. [2] Social, Genetic and Developmental Psychiatry Centre, Institute of Psychiatry, Psychology & Neuroscience, King's College London, London, UK. [3] Centre for Cognitive Ageing and Cognitive Epidemiology, University of Edinburgh, Edinburgh, UK. [4] Department of Psychology, University of Edinburgh, Edinburgh, UK. [153]These authors contributed equally: Heather C. Whalley, Andrew M. McIntosh. *A list of members and their affiliations are listed at the end of the paper. ✉email: andrew.mcintosh@ed.ac.uk

Major depression is the leading contributor to the global burden of disease[1], due to its high prevalence[2], disabling consequences[2] and partial treatment response[3]. Major depression is heritable ($h^2 = 37\%$)[4] and recent genome-wide association studies (GWAS) by Wray et al.[5] and Howard et al.[6] for the Psychiatric Genomics Consortium (PGC) have identified 44 and 102 risk-associated genetic variants, respectively. Although each single genetic variant contributes very little to disease liability, the genetic risk scores based on the additive effect of common genetic variants over the whole genome, i.e. polygenic risk scores (PRS), can account for a significant proportion of phenotypic variance[7]. The latest GWAS of depression now provides the ability to more precisely estimate polygenic risk of depression in independent samples[6] and thereby identify traits whose genetic architecture is shared with major depression using PRS.

Major depression is phenotypically correlated with many behaviours, brain structure and function measures, cognitive domains and physical conditions[8–13]. It is important to investigate the associations between the genetic predisposition to major depression and these phenotypes, to help identify shared causal risk factors, mechanisms and the causal consequences of major depression[14]. Until recently, however, this approach has received relatively little attention owing to a lack of data resources with the appropriate scale and coverage of genetic, behavioural and neuroimaging traits to test for these associations with sufficient statistical power[15–17].

A phenome-wide association study (PheWAS) aims to identify multiple phenotypes associated with a single genetic risk score or genotype. Unlike studies that examine associations between a single trait and genetic risk scores, PheWAS are less constrained by prior assumptions. This is particularly important in situations where we currently have an incomplete understanding of disease mechanisms. Genotype-based PheWAS approaches also have the considerable advantage that they are based on robust biological knowledge that is fixed from birth and therefore less susceptible to confounding and reverse causality.

The present study uses large data sets for both depression-PRS generation and a wide range of phenotypes, including neuroimaging. Depression-PRS are generated using summary statistics from the most recent meta-analysis combining the PGC, UK Biobank and 23andMe ($N = 0.8$ million)[6]. A PheWAS approach is used to estimate the effect and significance of associations between depression-PRS and other behavioural, cognitive and neuroimaging traits. A PheWAS is conducted on the latest neuroimaging data releases from the UK Biobank imaging project[18] that includes a discovery sample of 10,674 people, and a replication sample of 11,214 people (21,888 individuals in total). The UK Biobank imaging project is a large-scale data set containing both genetic and cross-modality neuroimaging data. A total of 77 traits are associated with polygenic risk of depression both in discovery and replication samples. Where depression-PRS are associated with neuroimaging phenotypes, we additionally test whether this is a potential causal consequence of depression or, conversely, whether neuroimaging measures have a causal effect on depression, using Mendelian Randomisation (MR) and structural equational modelling (SEM). The findings suggest that variation in white matter microstructure is a consequence of depression. We also test for the presence of gene-by-environment interactions using measures of early-life risk factors and socio-demographic variables available in UK Biobank[19,20], which show larger effects of polygenic risk of depression on psychiatric conditions when participants are exposed to adverse environments.

## Results

**PheWAS.** We found that 100 phenotypes (67 behavioural and 33 neuroimaging) out of 552 examined (209 behavioural and 343 neuroimaging) in the discovery sample showed significant associations with depression-PRS at a minimum of four $p$ thresholds after FDR (false discovery rate) correction for multiple comparisons (absolute $\beta$: 0.014–0.341, $\beta$ are standardised regression coefficients throughout, $p_{FDR}$ for linear regression: $0.050–3.61 \times 10^{-31}$). There were 37 phenotypes that remained significant after Bonferroni correction. However, due to correlation between the phenotypes tested, Bonferroni correction is likely to be overly conservative. Thresholds for both FDR and Bonferroni corrections are shown in Supplementary Fig. 1.

Overall results for depression-PRS of representative $p$ thresholds of 1 and 0.01 are presented in Fig. 1. These two thresholds were selected since pT < 1 and pT < 0.01 showed the largest effect sizes in behavioural traits and neuroimaging phenotypes, respectively (see Fig. 2 and Supplementary Fig. 1). Results for other thresholds can be found in Supplementary Fig. 1, Supplementary Table 1 and Supplementary Data 1-2.

All of the 100 variables showed an identical direction of effect in the replication sample (Fig. 3, Supplementary Figs. 2 and 3 and Supplementary Data 2 and 3). After multiple comparison correction, 77 traits showed associations with depression-PRS at a minimum of four $p$ thresholds in the replication sample (51 behavioural and 26 neuroimaging). Within these traits, 23 remained significant after Bonferroni correction. In total, 77% findings were replicated; the highest replication rates were seen for white matter microstructure (92.3%), mental health variables (81.3%) and physical measures (76%), see Supplementary Figs. 2 and 3. There was no significant interaction between magnetic resonance imaging (MRI) site and depression-PRS on any of the traits ($p_{cor} > 0.431$, see Supplementary Fig. 4, Supplementary Data 4). Results for meta-analysis combining the two samples can be found in Supplementary Figs. 5 and 6 and Supplementary Data 5.

Significant associations that were found in both the discovery and replication data sets are reported below; the $\beta$s provided are from the discovery analysis. A complete list of all results is presented in Supplementary Data 2 and 3.

For the associations between depression-PRS and definitions for depression and symptomology, higher depression-PRS were associated with the presence of depression based on all three definitions, including broad depression ($\beta$: 0.154–0.300, $p_{FDR}$ for linear regression: $3.93 \times 10^{-9}–3.61 \times 10^{-31}$), probable depression ($\beta$: 0.174–0.341, $p_{FDR}$ for linear regression: $1.14 \times 10^{-6}–1.52 \times 10^{-23}$), and Composite International Diagnostic Interview (CIDI) depression ($\beta$: 0.121–0.261, $p_{FDR}$ for linear regression: $3.08 \times 10^{-4}–1.18 \times 10^{-17}$). Significant associations were also found between depression-PRS and depressive symptoms, assessed by PHQ-4 (Patient Health Questionnaire) and CIDI questionnaires, and other self-reported psychological traits, including self-harm, subjective well-being, reported feeling of not worth living and neuroticism (absolute $\beta$: 0.027–0.339, $p_{FDR}$ for linear regression: $0.045–8.84 \times 10^{-30}$).

Associations were found between depression-PRS and white matter microstructure. Higher depression-PRS were in general associated with decreased white matter microstructural integrity. First, by looking at the classic microstructural measures of fractional anisotropy (FA) and mean diffusivity (MD): globally lower FA and higher MD (absolute $\beta$: 0.027–0.038, $p_{FDR}$ for linear regression: $0.037–6.90 \times 10^{-4}$) were associated with higher depression-PRS. Lower microstructural integrity was also found in the general measures of FA and MD for two subsets of white matter tracts, the association fibres (absolute $\beta$: 0.029–0.040, $p_{FDR}$ for linear regression: $0.024–6.05 \times 10^{-4}$) and thalamic radiations (absolute $\beta$: 0.025–0.036, $p_{FDR}$ for linear regression: $0.036–2.70 \times 10^{-3}$). For each individual tract (Figs. 2 and 4), higher depression-PRS were associated with decreased FA in inferior fronto-occipital

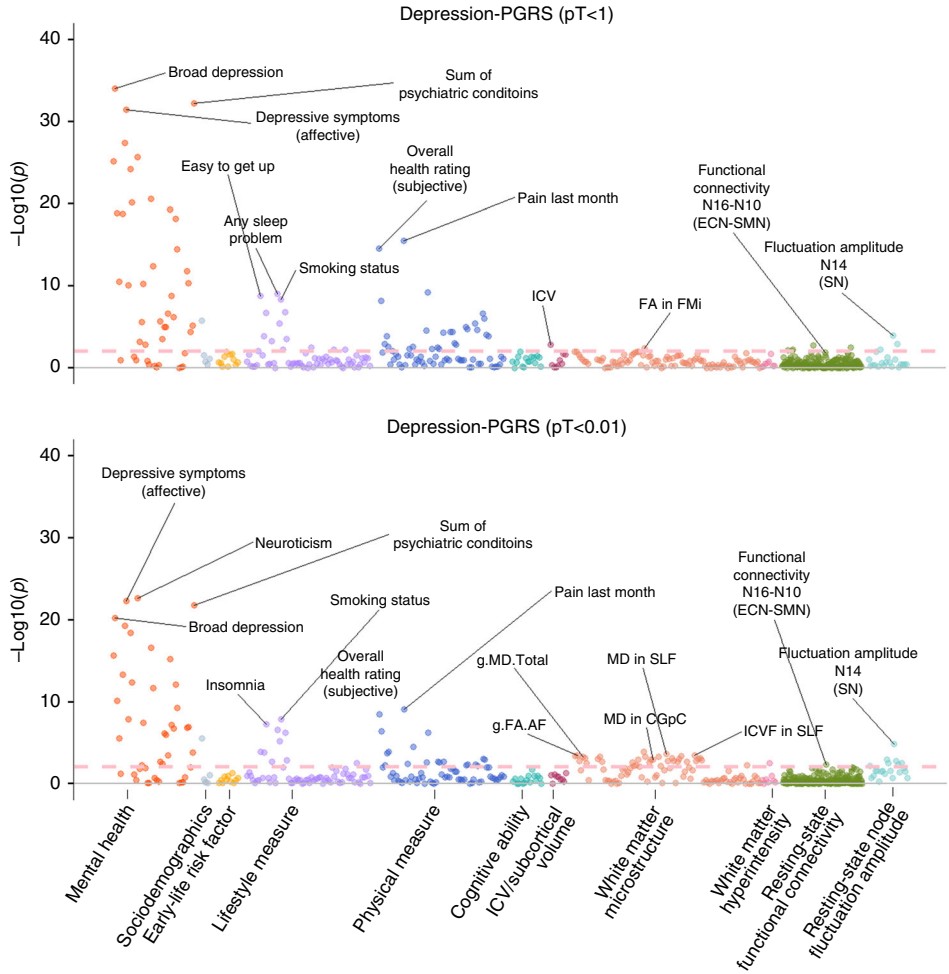

**Fig. 1 Significance plot for all phenotypes for depression-PRS at _p_ threshold (pT) < 1 and pT < 0.01.** The x axes represent phenotypes, and the y axes represent the $-\log_{10}$ of uncorrected _p_ values of two-sided test for linear regression between depression-PRS and each of the phenotype. Each dot represents one phenotype, and the colours indicate their according categories. The dashed lines indicate the threshold to survive FDR correction. FDR correction was applied over all the traits and all depression-PRS (see "Methods"). From left to right on the x axis, categories were shown by the sequence of mental health measure, sociodemographics, early-life risk factor, lifestyle measure, physical measure, cognitive ability, intracranial/subcortical volume, white matter microstructure, white matter hyperintensity, resting-state functional connectivity and resting-state fluctuation amplitude. Representative top findings are annotated in the figure. SN salience network, ECN executive control network, SMN sensorimotor network, FA fractional anisotropy, MD (for white matter microstructure) mean diffusivity, ICVF intra-cellular volume fraction, AF association fibres, FMi forceps minor, SLF superior longitudinal fasciculus, CGpC cingulate gyrus part of cingulum.

fasciculus, inferior longitudinal fasciculus, posterior thalamic radiation and superior longitudinal fasciculus (SLF) ($\beta$: $-0.025$ to $-0.032$, $p_{FDR}$ for linear regression: $0.050-5.78 \times 10^{-3}$) and increased MD in anterior thalamic radiation (ATR), cingulate gyrus part of cingulum, inferior fronto-occipital fasciculus, SLF and superior thalamic radiation ($\beta$: $0.023-0.042$, $p_{FDR}$ for linear regression: $0.040-6.95 \times 10^{-5}$). All associations found for neurite orientation dispersion and density imaging measures were in intra-cellular volume fraction (ICVF; an index reflecting neurite density, $\beta$: $-0.025$ to $-0.044$, $p_{FDR}$ for linear regression: $0.047-1.74 \times 10^{-4}$). General variance of ICVF in the association fibre subset was negatively associated with depression-PRS ($\beta$: $-0.028$ to $-0.039$, $p_{FDR}$ for linear regression: $0.036-1.13 \times 10^{-3}$). For tracts, lower ICVF was correlated with higher depression-PRS in similar regions found for FA and MD, in acoustic radiation, cingulate gyrus part of cingulum, inferior fronto-occipital fasciculus, SLF and uncinate fasciculus ($\beta$: $-0.025$ to $-0.043$, $p_{FDR}$ for linear regression: $0.043-2.10 \times 10^{-4}$).

Depression-PRS were also found associated with resting-state fluctuation amplitude. Associations were found between depression-PRS and resting-state fluctuation amplitude of low-frequency signal ($\beta$: $0.027-0.043$, $p_{FDR}$: $0.037-2.03 \times 10^{-4}$) in the discovery sample (Figs. 2 and 4). A full list of report is presented in Supplementary Data 2.

In brief, higher depression-PRS were associated with lower fluctuation amplitude in anterior cingulate gyrus (peak coordination: $-10$, $54$, $2$; cluster size: 7065), bilateral postcentral gyrus (peak coordination: $-44$, $-30$, $46$ and $44$, $-24$, $40$ for left and right hemispheres, respectively; cluster sizes: 2781 and 1619), bilateral insula (peak coordination: $-38$, $-4$, $16$ and $30$, $18$, $-16$ for left and right hemispheres, respectively; cluster sizes: 963 and 308), bilateral orbital part of inferior frontal gyrus (peak coordination: $-34$, $34$, $-12$ and $32$, $36$, $-10$ for left and right hemispheres, respectively; cluster sizes: 154 and 171) and left superior frontal lobe (peak coordination: $-18$, $32$, $38$; cluster size: 124). These regions are largely contained within the salience, executive control and sensorimotor networks (Supplementary Table 2)[15,21].

Finally, depression-PRS were found associated with sleep problems, smoking and poor physical health. In the category of

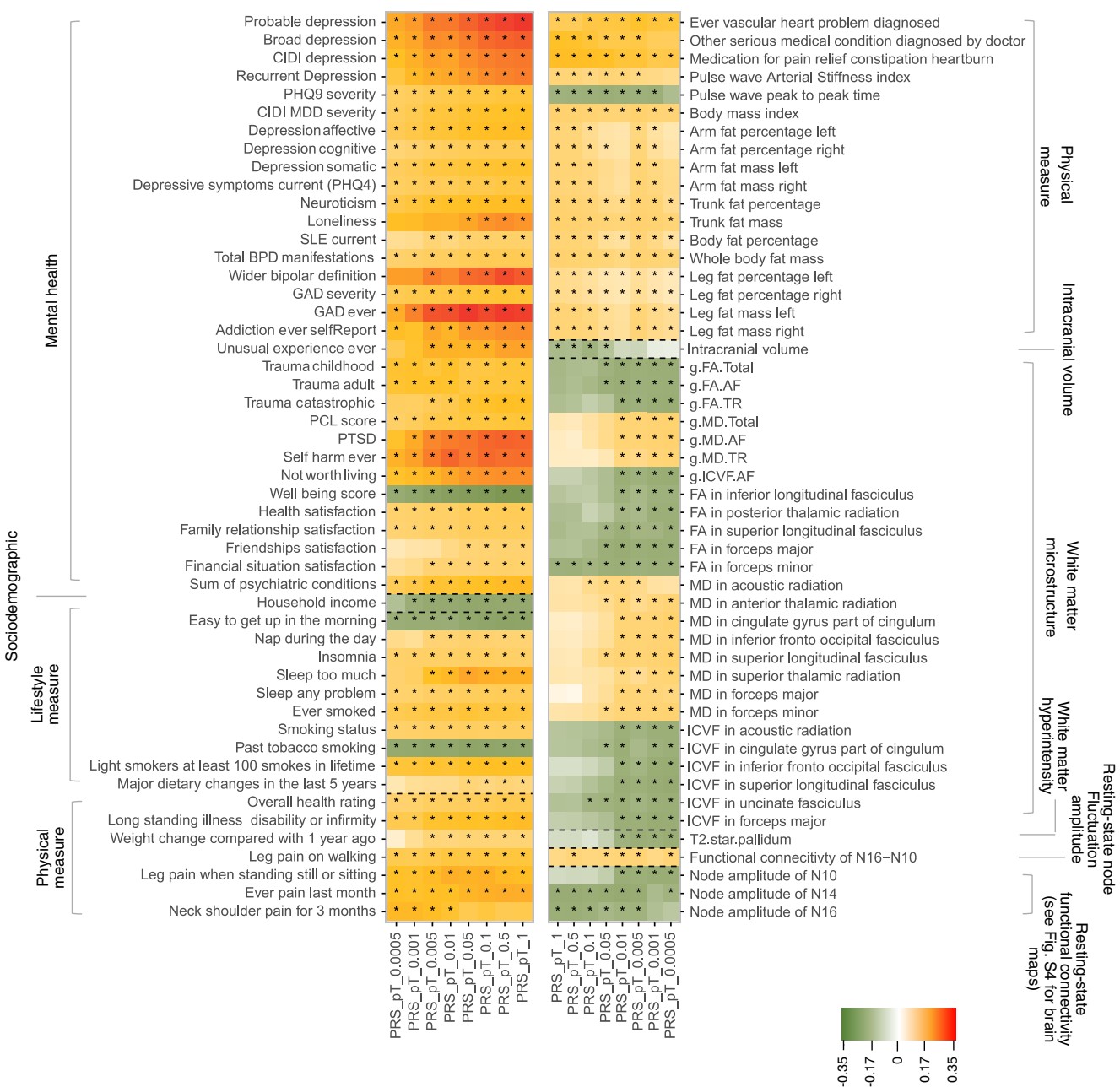

**Fig. 2 Heatmap for the traits that were significantly associated with depression-PRS.** The shown traits were significantly associated with depression-PRS at a minimum of four p thresholds for depression-PRS. Shades of cells indicate the standardised effect sizes (β) for the linear regression between depression-PRS and each phenotype. A larger effect size was shown by a darker colour. Cells with an asterisk were significant after FDR correction. Descriptions for the variables in detail can be found in Table 1, Supplementary Table 1 and Supplementary Data 1.

lifestyle measures, reporting of sleep problems (e.g. too much sleep or insomnia) (absolute β: 0.034–0.180, $p_{FDR}$ for linear regression: 0.043–8.26 × $10^{-9}$) and smoking behaviours (absolute β: 0.044–0.105, $p_{FDR}$ for linear regression: 2.28 × $10^{-3}$–3.74 × $10^{-8}$) were found to be significantly positively associated with depression-PRS.

Physical health items associated with depression-PRS can be summarised as the following four categories: (1) self-reported overall health rating and conditions of long-standing illnesses (absolute β: 0.040–0.129, $p_{FDR}$ for linear regression: 4.38 × $10^{-3}$–1.49 × $10^{-13}$), (2) recent pains and on-going treatment for pain (absolute β: 0.083–0.163, $p_{FDR}$ for linear regression: 6.30 × $10^{-4}$–1.28 × $10^{-14}$), (3) cardiovascular/heart problems (absolute β: 0.066–0.112, $p_{FDR}$ for linear regression: 0.027–1.97 × $10^{-5}$), and (4) body mass and weight

change compared to 1 year ago (absolute β: 0.014–0.042, $p_{FDR}$ for linear regression: 0.046–5.00 × $10^{-6}$).

**Bidirectional MR on imaging phenotypes and depression.** A significant and potentially causal effect of depression was found on lower microstructural integrity in four white matter microstructural measures and lower resting-state fluctuation amplitude in the Salience Network (Node 14). For these phenotypes, the effect from depression were shown using at least two MR methods after FDR correction (Fig. 5, Supplementary Data 6 and 7 and Supplementary Figs. 7–11). The neuroimaging phenotypes include (β and $p_{FDR}$ reported for significant effects): global gMD (gMD-Total; β: 0.125–0.724, $p_{FDR}$ for MR: 0.041–0.022, significant for all three MR methods), gMD in thalamic radiations

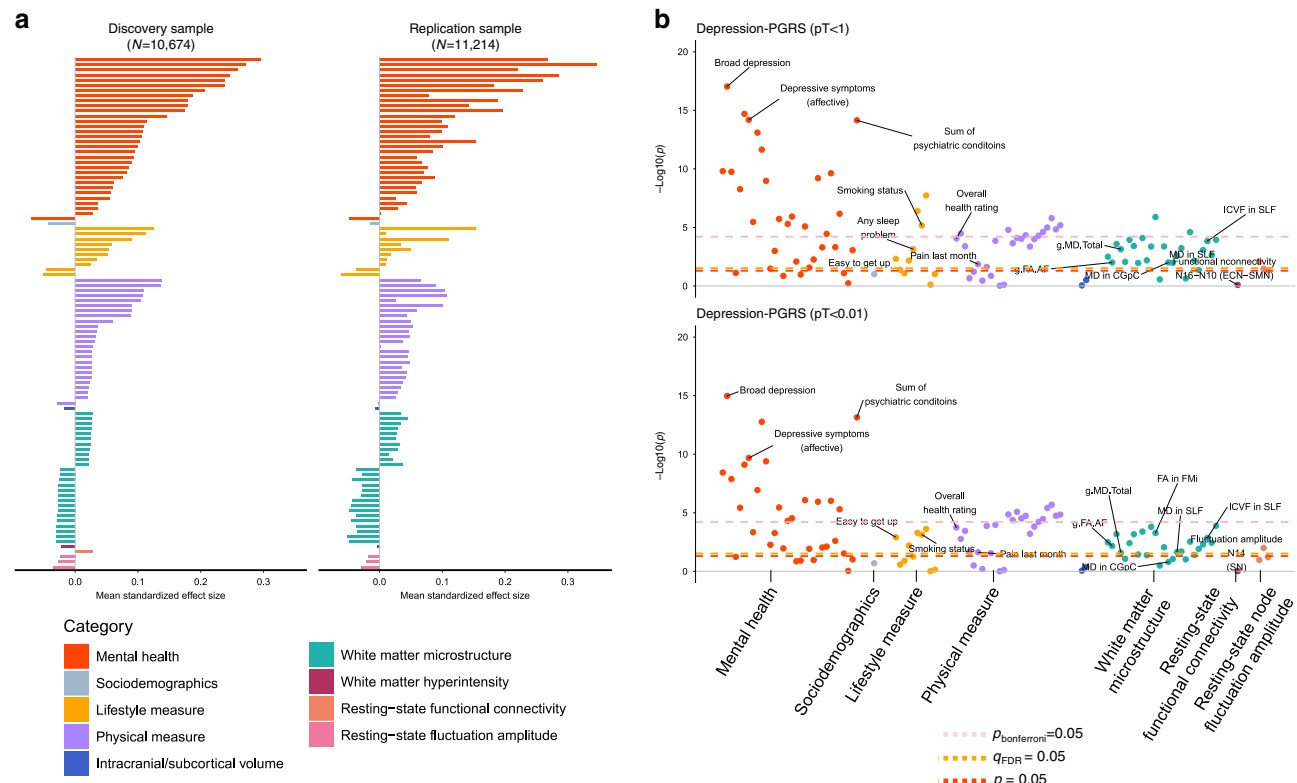

**Fig. 3 Results for replication analysis. a** Comparisons of effect sizes for the discovery and replication samples. The *x* axes represent the mean standard effect size across depression-PRS at all eight *p* thresholds for generating depression-PRS (pT). Colours for the bars indicate their categories (from top to bottom: mental health measure, sociodemographics, lifestyle measure, physical measure, intracranial/subcortical volume, white matter microstructure, white matter hyperintensity, resting-state functional connectivity, and resting-state fluctuation amplitude). **b** Significance plot for the replication analysis on representative depression-PRS at pT < 1 and pT < 0.01, in accordance with Fig. 1. The *x* axes represent phenotypes, and the *y* axes represent the $-\log_{10}$ of uncorrected *p* values of two-sided test for linear regression between depression-PRS and each of the phenotype. Each dot represents one phenotype, and the colours indicate their according categories. The yellow dashed lines indicate the threshold to survive FDR-correction. FDR-correction was applied over all the traits and all depression-PRS (see "Methods"). The pink and red dashed lines indicate the threshold to survive Bonferroni correction and nominally significant threshold. Top hits shown in the discovery sample (Fig. 1) are annotated in the figure. Explanations for the abbreviations can be found in the legend of Fig. 1.

(gMD-TR; $\beta$: 0.131–0.527, $p_{FDR}$ for MR: 0.050–0.010, significant for all three MR methods), ICVF in SLF (tICVF-SLF; $\beta$: −0.159 to −0.926, $p_{FDR}$ for MR: 0.023–0.015, inverse-variance weighted estimator (IVW) and MR-Egger) and forceps major (tICVF-FMa; $\beta$: −0.160 to −0.792, $p_{FDR}$ for MR: 0.040–0.023, IVW and MR-Egger), and the resting-state fluctuation amplitude in the Salience Network (amp-N14; $\beta$: −0.130 to −0.177, $p_{FDR}$ for MR: 0.021–0.015, IVW and the weighted median). Other than ICVF in SLF, no significant reverse effects of these neuroimaging phenotypes on depression were found (*p* for MR ranged from 0.860 to 0.498). For the above significant effects, ICVF in SLF and FMa both showed significant heterogeneity among genetic instruments, indicating potential horizontal pleiotropy ($p_{FDR}$ for Q test: 0.018–0.011, for MR-Presso global test: 0.024–0.009), and after removing outlying genetic instruments, MR-Presso became insignificant for both variables ($\beta$: −0.086 to −0.087, *p* for MR: 0.090–0.081). No other test showed significant horizontal pleiotropy or single-nucleotide polymorphism (SNP) heterogeneity ($p_{FDR}$ for MR-Egger intercept > 0.071, $p_{FDR}$ for all Q tests > 0.135 and $p_{FDR}$ for MR-Presso global test > 0.214).

Conversely, the directional effect of neuroimaging phenotypes on depression was then tested. The only significant effect was shown from general variance of ICVF in association fibres to depression for IVW method (gICVF-AF; $\beta$ = −0.031, $p_{FDR}$ for MR = 0.018); however, results using other MR methods were not significant ($p_{FDR}$ for MR > 0.886). Heterogeneity tests were also

highly significant ($p_{FDR} = 4.78 \times 10^{-7}$ for Q test and $3.33 \times 10^{-4}$ for MR-Presso global test). Two other neuroimaging phenotypes showed nominally significant effects on depression, including higher MD in the ATR for MR-Egger (tMD-ATR; $\beta = 0.107$, *p* for MR-Egger = 0.028, $p_{FDR} = 0.166$; however, the Egger intercept was not significant, $p_{FDR} = 0.432$, Supplementary Fig. 11, Supplementary Data 6 and Supplementary Table 3) and ICVF in SLF (tICVF-SLF; $\beta = −0.025$, *p* = 0.030, $p_{FDR} = 0.179$ for IVW).

An additional test was conducted to see whether there was a substantial reduction in effect sizes after controlling for depressive symptoms (assessed online by CIDI short form and PHQ-9, and PHQ-4 for current symptoms along with the imaging assessment), and all three white matter microstructural measures were found significant in the MR analysis as the causal consequences of depression showed large reductions in effect sizes (reduced by 20.5–30.9%), however, resting-state fluctuation amplitude in Salience Network did not show such a reduction (by 0.3%, see Supplementary Figs. 12–14).

In addition to the above MR results, genetic correlations were found between depression and FA in forceps minor (rg = −0.157, $p_{cor}$ for genetic correlation = 0.001), MD in ATR (rg = 0.106, $p_{cor}$ for genetic correlation = 0.012), MD in cingulate part of cingulum (rg = 0.105, $p_{cor}$ for genetic correlation = 0.012), MD in forceps minor (rg = 0.119, $p_{cor}$ for genetic correlation = 0.012), general ICVF in association fibres (rg = −0.083, $p_{cor}$ for genetic correlation = 0.026), ICVF in cingulate part of cingulum (rg =

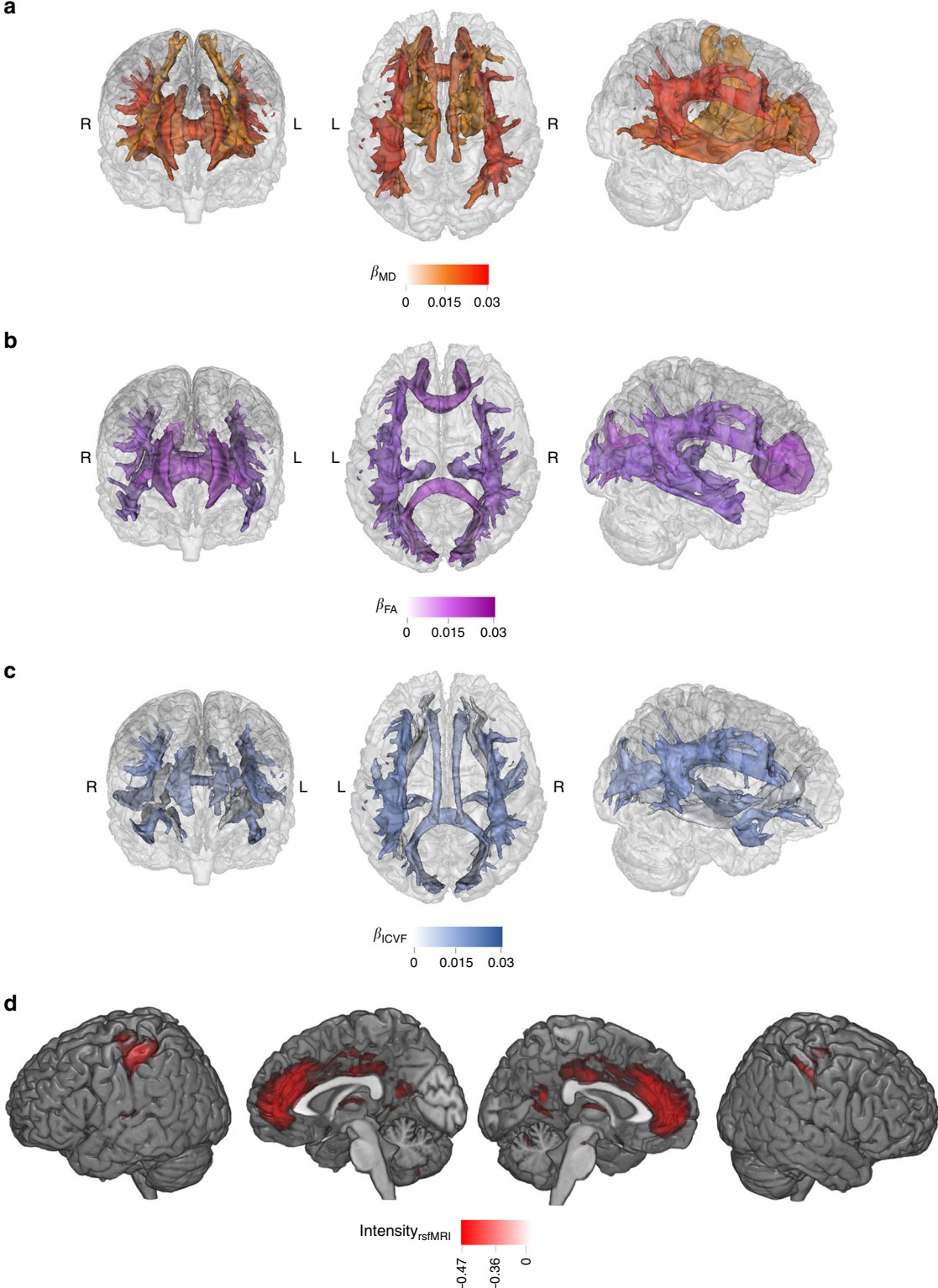

**Fig. 4 Maps for the significant associations between depression-PRS and brain phenotypes. a–c** are the brain maps for the significant associations between depression-PRS and white matter microstructure in fractional anisotropy (FA; **a**), mean diffusivity (MD; **b**) and intra-cellular volume fraction (ICVF; **c**) of major tracts. The shade for each tract represents the standardised effect size ($\beta$), with a darker shade showing a greater mean $\beta$ across all depression-PRS at different $p$ thresholds (pT). From left to right are from anterior, superior and right view. For clarity, among the tracts presented in Fig. 2, the ones that showed consistent associations across at least four depression-PRS pT are presented. **d** shows the brain maps for regions involved in significant associations between resting-state fluctuation amplitude and depression-PRS. Regions that show consistent associations across at a minimum of four depression-PRS $p$ thresholds are presented. Visualisation of results is achieved by calculating the average intensity of ICA maps, weighted by their mean $\beta$ across the pT. For clarity, the brain maps shown below have a threshold applied on (intensity over 50% of the highest global intensity).

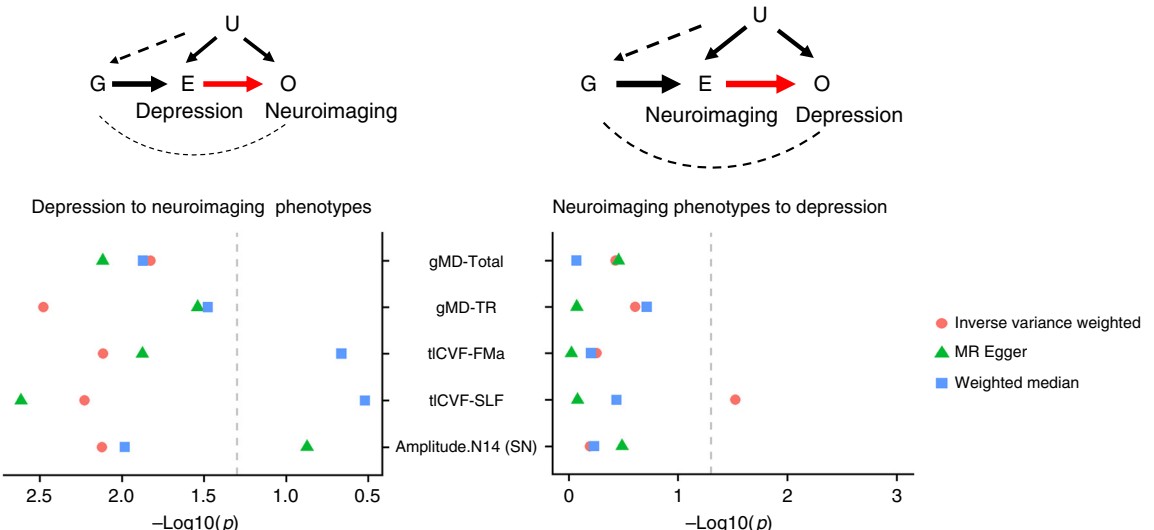

**Fig. 5 Mendelian Randomisation analysis between neuroimaging phenotypes and depression.** The left panel shows the model and results for Mendelian Randomisation results for the causal effect of depression to neuroimaging phenotypes, and the right panel shows the model and results for effect of neuroimaging phenotypes to depression. For the model illustrations, G = genetic instruments extracted from GWAS summary statistics of the exposure, E = exposure variable, O = outcome variable, U = unmeasured confounders (have no systematic association with G). In the scatter plots, x axes represent −log10-transformed p values for the Mendelian Randomisation results, and the y axes represent the neuroimaging traits tested in the models. Three types of dots represent the three Mendelian Randomisation methods used. Dashed grey lines are the p = 0.05 threshold for nominal significance. MD mean diffusivity, ICVF intra-cellular volume fraction, TR thalamic radiations, SLF superior longitudinal fasciculus, Amplitude.N14 (SN) fluctuation amplitude in Node 14 (i.e. the Salience Network).

−0.10, $p_{cor}$ for genetic correlation = 0.023), SLF (rg = −0.10, $p_{cor}$ for genetic correlation = 0.023) and uncinate fasciculus (rg = −0.10, $p_{cor}$ for genetic correlation = 0.023) (see Supplementary Table 4).

**Mediation analyses on imaging phenotypes.** In the first mediation model, we tested whether polygenic risk of depression led to changes in several neuroimaging variables through the mediating effects of depression. The neuroimaging variables were chosen if they presented as a significant causal consequence of depression in the MR analyses. Conversely, in the second model the neuroimaging variable of MD in ATR showed a potentially causal effect on depression at nominal significance using MR and was therefore tested for its potential role as a mediator of genetic risk on depression. Other neuroimaging variables nominally significant in the MR analysis as causal factors were not tested as mediators, because the heterogeneity tests were highly significant. Here we report the results for depression-PRS at the threshold of pT < 1. For other depression-PRS thresholds, see Supplementary Data 8. Details can be found in Supplementary Methods.

We found evidence that current depressive symptoms measured by PHQ-4 mediated the effect of depression-PRS on: global MD (gMD-Total; $\beta = 0.002$, $p_{FDR}$ for mediation test = 0.003) and MD in thalamic radiations (gMD-TR; $\beta = 0.002$, $p_{FDR}$ for mediation test = 0.001). Conversely, a significant mediation effect of MD in ATR was found, mediating the effect of depression-PRS on current depressive symptoms (PHQ-4) ($\beta = 0.001$, $p$ for mediation test = 0.005). All significant mediation models showed good model fit characteristics (CLI ranged from 0.987 to 0.993, TLI ranged from 0.978 to 0.989, and all $p_{RMSEA} = 1$). A full list of results for all mediation models tested can be found in Supplementary Data 8.

**G-by-E interaction.** Environmental variables that showed significant interaction with depression-PRS included childhood trauma, Townsend Index and recent stressful life events. The dependent variables that provided evidence of G × E were mainly measures of mental health, including depressive symptoms and the self-declared total number of psychiatric conditions (see Fig. 6 and Supplementary Fig. 15, $p_{FDR}$ for linear regression <0.040).

In general, the effect of depression-PRS was enhanced in participants exposed to more adverse social/socioeconomic environments. In participants who reported any childhood trauma versus none, the variance in the dependent variables accounted for by depression-PRS were 1.67–1.78 times higher for the total number of psychiatric conditions and affective symptoms of depression. For participants in the most deprived tertile band, variance explained in the sum of psychiatric conditions was 3.57 times higher than for the least deprived participants. Detailed reports can be found in Fig. 6, Supplementary Fig. 15 and Supplementary Data 9–14.

We found no evidence of interactions however between depression-PRS and adulthood trauma, recent stressful life events and household income ($p_{FDR}$ for linear regression >0.086).

## Discussion

Replicated associations between depression-PRS, behavioural and neuroimaging phenotypes were found in the present study using an independent imaging cohort. The strongest associations were found between depression-PRS and mental health variables. Several novel associations were detected, including associations between depression-PRS and both brain white matter microstructure and a measure of resting-state activity amplitude. In addition, MR analysis also showed evidence for changes in the MD of thalamic radiations and global variance of MD that, should the assumptions of MR hold, are likely to be a causal consequence of depression. Other associations with higher polygenic risk included more abnormal self-reported sleep problems, smoking behaviour and presence of cardiovascular conditions, as well as an increased in body mass index. Findings regarding the interactions of early-life factors and sociodemographic variables with depression-PRS revealed that the effect of depression-PRS

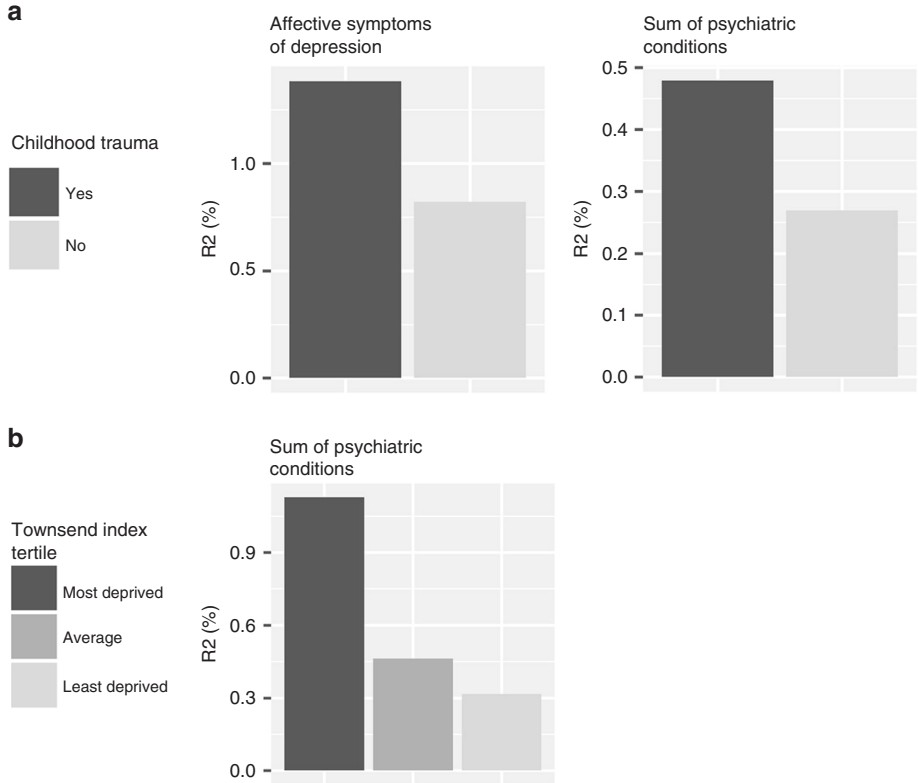

**Fig. 6 G-by-E interaction.** The figures present the variance explained by depression-PRS under the exposure of different environmental risk factors. The colour shade of each bar represents one condition of environmental factor, a darker shade represents a risk-conferring condition (i.e. had reported childhood trauma and in the most deprived area). The y axes represent the variance explained ($R^2$ in %) by depression-PRS under the given environmental conditions.

on mental health was stronger in participants who reported childhood trauma and experienced socioeconomic deprivation.

While replicated associations were found between depression-PRS and both behavioural and neuroimaging variables, in total 24.4% of all the behavioural phenotypes tested were found to be significantly associated with depression-PRS. The proportion was lower for the neuroimaging phenotypes tested, where only 7.6% of the variables were significantly associated. The higher proportion of associations found for behavioural phenotypes likely reflects the overlapping genetic architecture of multiple psychiatric conditions with one another and the behavioural traits with which they are commonly associated. In contrast, although several brain phenotypes were associated with depression PRS, our findings suggest that depression shares its genetic architecture with only a small proportion of them. One potential reason for this finding is a relative lack of signal in the GWAS of neuroimaging variables[22]. It is also possible that depression may have a relatively specific relationship with a smaller number of neuroimaging variables, reflecting the underlying mechanisms of depression.

Novel associations were found between depression-PRS and neuroimaging variables on structural connectivity and functional resting-state fluctuation amplitude in the brain. Findings from both diffusion tensor imaging and resting-state data revealed the importance of the prefrontal cortex, which is a hub for emotion regulation and executive control[23,24]. The role of the prefrontal cortex is further supported by a GWAS on depression in UK Biobank, which showed the enrichment of risk-associated genes in this region[25]. In particular, white matter microstructure showed the largest effect sizes among brain phenotypes in our results, and most trait associations in this category were replicated

in an independent data set. The current findings therefore indicate a potentially risk-conferring role for white matter over other modalities. This finding is supported by previous evidence that white matter microstructure has stronger phenotypic associations with lifetime depression compared to brain structural volumes[26] and higher SNP heritability (20–60%) compared with other neuroimaging modalities, indicating a greater genomic contribution to individual differences in phenotypes[27]. Resting-state findings were comparatively less replicated than for white matter, which may be due to the less well-standardised protocols for resting-state acquisition. For example, although others have shown that the results of resting-state studies are broadly comparable across a range of acquisition lengths[28], the data acquisition time for resting-state data in UK Biobank was relatively short compared with other imaging cohorts, such as the Human Connectome Project[29].

The strongest replicated white matter finding with PRS were found for the MD measures, consistent with previously reported depressive symptom associations[30]. This may due to MD's greater sensitivity to ageing and related pathophysiological processes in this mid- to late-life UK Biobank sample[31]. Alternatively, the associations with dispersion density suggest that reductions in MD may be partly due to reduced neurite density. This highlights the need for further investigation of these issues in tissue from large samples of depressed individuals. Recent gene expression studies suggest that genetic predisposition to depression may influence more spatially and functionally specific, neuronal-level activities such as synaptic pruning and the overproduction of synapses[32] for regional segregation[33] during the process of brain maturation and myelin repair, which contribute largely to brain structural and functional individual variance[27]. These highly

regional and functionally specific brain phenotypes are of great importance and may help explain how genetic predisposition contributes to variance in neuroimaging measures.

Several associations between polygenic risk of depression and neuroimaging variables were subsequently identified, through MR analysis, to have directional or potential causal significance. Whether brain structural and functional alterations are the outcome or cause of depressive symptoms has long been debated[34]. Our results show that some brain structural and functional alterations are likely to be an outcome of depression; however, whether other imaging features are also a cause is yet unclear. Although our results for the causal effect from neuroimaging phenotype to depression were null, therefore suggesting a possibly uni-directional relationship from depression to the brain, it may be premature to draw confident conclusions without the availability of a greater number of genome-wide significant genetic instruments for neuroimaging traits. It is important to consider that the relative lack of genome-wide significant loci for most neuroimaging measures provides weaker genetic instruments for MR, which may reduce power to detect causal associations. There is currently a global effort to conduct GWAS using neuroimaging phenotypes, and these efforts are likely to provide stronger genetic instruments for future analyses. Further, white matter microstructure in ATR did demonstrate a nominally significant causal effect on depression, but notably not in the reverse direction (from depression to ATR). This is in spite of the reverse direction of testing (from depression to ATR) having a much larger set of genetic instruments and greater power to detect significant effects. This indicates that the white matter microstructure in ATR may be one of the strongest neuroimaging candidates as a causal mediator of risk for depression.

The associations found in behavioural traits with depression-PRS suggest that polygenic risk of depression may also identify a predisposition to experience particular environmental risk exposures, or a vulnerability to their effects and later recall. First, the linear association of depression-PRS with sleep, recent pains, smoking behaviour and the presence of any heart/cardiovascular conditions showed the largest effect sizes. Various mechanisms can be involved in these behavioural patterns, such as hyperactivity in the hypothalamic–pituitary–adrenal (HPA) axis[35], and neurodevelopmental or parental impact on poorer health. Studies have shown that poor physical and neurobiological health may be correlated[36]. Here we identified candidate behavioural and physical phenotypes that may partially explain the genetic association between depression and brain phenotypes for future research to explore. Second and more directly, the environmental risk factors tested in this study consistently strengthened the effect of depression-PRS. Compared with previous studies that test genetic–environment (G × E) interactions, the present study revealed that the G × E effect can present on a whole-genome, polygenic level. It may be a manifestation of interactions between the environmental risk factors and some important endophenotypes (e.g. HPA-axis activity) that polygenic risk of depression confers upon.

There are several strengths and limitations to the present study. PheWAS aims to identify the multiple phenotypes associated with a single risk score. This is arguably a stronger approach than studies that consider a single trait, based on prior theory, as PheWAS is less constrained by prior assumptions based on an incomplete understanding of disease mechanisms. Genotype-based PheWAS approaches also have the considerable advantage that they are based on robust biological knowledge that is fixed from birth and less susceptible to reverse causality. The current study leveraged the most up-to-date GWAS findings in depression, providing the most predictive PRS for depression with >100 instruments to test for bi-directional causal associations with

brain imaging variables, and improved power for the detection for significant G × E interactions. This approach led to a number of novel findings, including MR-based evidence for a causal effect of depression on measures of brain function and connectivity. The current data set has considerable advantages in terms of its large sample size and was focussed on whether polygenic associations can be replicated across neuroimaging samples, improving on an area of previously identified methodological weakness[37]. Most neuroimaging studies have sample sizes in the range of 50–100 people[38]. In contrast, the current study provided results from a larger sample using a potentially less biased, data-driven approach.

The present study uses MR to address causal relationships between depression and brain imaging measures. The directional or causal relationship between these traits has remained uncertain. Our approach takes a more methodologically consistent approach and applies state-of-the-art causal inference methods to go beyond mere association, prioritising brain regions and risk factors for experimental approaches.

Although we found robust and replicated associations between brain measures and genetic risk of depression, whether neurodevelopmental or neurodegenerative factors are both contributing to the individual differences is unknown. Participants in this study were in their mid-life to later life. Various factors, including ageing[39], the long-term effects of early developmental deficits[40] and comorbid illnesses, may impact variation in brain phenotypes in this age range. These possible explanations necessitate longitudinal imaging data and studies of high-risk participants that are able to identify the timing and trajectory of brain differences before and after the onset of illness. The genomic regions driving the shared architecture between depression and the brain phenotypes have also not been identified.

The mediation models employed in our investigation were limited in that they tested for causal associations using cross-sectional, rather than longitudinal, data. In order to make causal inferences, we sought consistent findings using both of these methodologies but acknowledge that other methodologies will help to facilitate more robust causal inferences in future. Larger samples for genetic studies on neuroimaging traits would largely benefit such analysis in order to balance the statistic power of clinical and neuroimaging phenotypes.

Future studies that provide improved GWAS on depression and relevant traits would further increase our understandings of depression. Summary statistics we used here were based on GWAS that included some cases identified by self-declared depressive symptoms. As it has been argued in previous papers, the self-declared phenotypes may, to some extent, be more lenient than clinically identified traits; however, the statistic power can largely overcome the noise introduced by a small amount of misclassification, which was supported by a high genetic correlation between self-declared depression and clinically validated depression[5,6]. While PRS is a powerful means of identifying factors associated with genetic risk, it currently explains around 1.6% of phenotypic variance in depression. Future PRS scores, trained on more precise GWAS summary statistics, are likely to be more strongly predictive and may have greater sensitivity to detect disease-relevant phenotypes. Further associations may be revealed as PheWAS studies increase in size, although this is counterbalanced by their small effect sizes and likely limited clinical utility for individual patients.

To conclude, a novel and relatively unconstrained approach was used to test for associations between depression-PRS and various behavioural and neuroimaging variables of likely relevance for depression. The findings revealed that white matter microstructure, general mental and physical health and behaviours such as sleep patterns and smoking behaviour were

associated with PRS of depression. Our findings suggest that most neuroimaging associations with depression are likely to be the causal consequence of depression.

## Methods

**Participants**. Data from 21,888 individuals who participated in the UK Biobank imaging study[18] were included in the current study (released in 2 waves, in May and October 2018, mean age is 62.75 years, standard deviation of age is 7.44 years, 48.4% were male, details can be found in Supplementary Table 5). The discovery sample included participants mainly from the first data release, and the replication sample from the second release (details for the discovery and replication samples can be found in Supplementary Fig. 16). The majority of participants were assessed at the Cheadle MRI site (80.1%) and the rest at the Newcastle site (19.9%). All imaging data were collected using a 3-T Siemens Skyra (software platform VD13) machine.

Behavioural and neuroimaging data acquisition were conducted under standard protocols[18,41]. Written consent was acquired for all participants. Data acquisition and analyses in the present study were conducted under UK Biobank Application #4844. Ethical approval was accepted by the National Health Service (NHS) Research Ethics Service (11/NW/0382).

**Depression-PRS**. In the present study, the sample used for generating GWAS summary statistics is referred to as the training data set. The samples in which depression-PRS were generated and tested are referred to as the testing samples, which include both discovery and replication samples (as described above). We removed any overlapping individuals from the training sample (used to estimate allele effects for polygenic profiling) and testing data sets (where the effects of PRS scores were estimated) (see Supplementary Methods).

PRS were calculated using the summary statistics from a meta-analysis of depression GWAS from three cohorts, including PGC analysis of major depression[5], the 23andMe discovery sample in the Hyde et al. analysis of self-reported clinical depression[42] and a broad depression phenotype from UK Biobank within individuals who had not participated in the imaging study[25]. This meta-analysis provided a total training data set of 785,581 individuals (238,360 cases and 547,221 controls; for further details, see the study by Howard et al.[6]). We used the summary statistics that included only the 8,099,819 SNPs that were present in the GWAS data from all three cohorts[6].

PRSice version 2.0 (used with PLINK 1.9)[43] was used to calculate the depression-PRS. Before the analyses were conducted, individuals who met the following criteria were removed from the testing data set: related or non-European-ancestry individuals and those who were included in PGC, 23andMe and UK Biobank GWAS on depression (details can be found in Supplementary Methods). The sample sizes reported below are after applying the above criteria. Genotyping and quality control were conducted by UK Biobank as described in an earlier protocol paper[44]. Details of SNP quality control and imputation can be found in Supplementary Methods. We used the classic thresholding+clumping method to generate PRSs. This method allows direct comparisons with a vast majority of previous major depressive disorder (MDD)-PRS studies that used the same approach. We did not consider some new Bayesian methods because they showed no particular advantages over the thresholding+clumping method for MDD[45]. Eight $p$ value thresholds were applied to select genetic variants included in calculating PRS, as $p < 0.0005$, $p < 0.001$, $p < 0.005$, $p < 0.01$, $p < 0.05$, $p < 0.1$, $p < 0.5$ and $p < 1$.

**Behavioural phenotypes**. The behavioural phenotypes consisted of 6 broad categories, containing 209 variables in total. Where summary data were available (e.g. neuroticism total score), the individual items used to derive the summary data were not included. Phenotypes that were available on <2000 people in the discovery sample were also excluded from further analysis. Mean sample sizes for all traits contained in each category are included in brackets below. For further details see in Table 1, Supplementary Table 1 and Supplementary Data 1. Categories included: (1) Mental health ($N_{discovery} = 7970$ and $N_{replication} = 3880$), including self-reported symptoms of major psychiatric conditions[46]. In this category, three definitions for depression were included: broad depression, which was a self-declared definition of whether the participant had seen a psychiatrist for nerves, anxiety, tension or depression[6,25], probable depression which was derived from an abbreviated set of self-declared symptoms of major depression and hospital admission history[47], and CIDI depression, a measure assessing full diagnostic criteria for depression based on questions from a shortened version of the structured CIDI[46]. (2) Socio-demographic measures ($N_{discovery} = 8759$ and $N_{replication} = 4352$), such as house-hold income and educational attainment. (3) Early-life risk factors ($N_{discovery} = 9755$ and $N_{replication} = 10,370$), containing physical measures such as birth weight, and environmental variables like adoption and maternal smoking. (4) Lifestyle measures ($N_{discovery} = 9231$ and $N_{replication} = 4796$), which mainly included items on sleep, smoking, alcohol consumption and diet, (5) Physical measures ($N_{discovery} = 8961$ and $N_{replication} = 4618$), consisting of self-declared medical conditions such as recent pains, cancers, operations, heart and artery diseases and other major illnesses and also measures of blood pressure, arterial stiffness and hand-grip strength, and finally (6) Cognitive ability ($N_{discovery} = 8153$ and $N_{replication} = 4105$).

## Table 1 A summary of phenotypes.

| Category | Number of traits | N for Discovery sample | N for Replication sample | UK Biobank data modality |
|---|---|---|---|---|
| Mental health | 44 | Range: 3299–10,674; Mean: 7970 | Range: 1519–5565; Mean: 3880 | Touchscreen, Online follow-up |
| Sociodemographic | 5 | Range: 8054–9941; Mean: 8759 | Range: 3824–5199; Mean: 4352 | Touchscreen |
| Early-life risk factor | 11 | Range: 7428–10,674; Mean: 9775 | Range: 8255–11,214; Mean: 10,370 | Touchscreen, Online follow-up |
| Lifestyle measures | 66 | Range: 2789–10,674; Mean: 9231 | Range: 1392–5565; Mean: 4796 | Touchscreen |
| Physical measures | 67 | Range: 2155–10,674; Mean: 8961 | Range: 1047–5565; Mean: 4618 | Touchscreen |
| Cognitive ability | 16 | Range: 5250–10,674; Mean: 8153 | Range: 2586–5565; Mean: 4105 | Touchscreen, Online follow-up |
| Intracranial/subcortical volume | 9 | Range: 10,627–10,631; Mean: 10,631 | Range: 5550–5553; Mean: 5553 | Brain imaging |
| White matter hyperintensity | 8 | 9702 | 5187 | Brain imaging |
| White matter microstructure | 95 | Range: 9341–9396, Mean: 9377 | Range: 5187–5261; Mean: 5239 | Brain imaging |
| Resting-state functional connectivity | 210 | 9745 | 5241 | Brain imaging |
| Resting-state fluctuation amplitude | 21 | 9745 | 5241 | Brain imaging |

A total of 209 behavioural phenotypes (6 categories) and 343 neuroimaging variables (4 categories) are included. Where there are multiple sample sizes in a category, a range and the mean of sample sizes (N) are presented.

This included four tests conducted at the assessment centres, four tests conducted online and a general measure[46] derived based on the tests conducted at the assessment centres that have larger sample sizes (see more details in Supplementary Methods).

All of the behavioural phenotypes, with the exception of mental health items derived from online follow-up questionnaires (see Table 1), were primarily acquired at the same time as the imaging assessment. Missing data for the imaging assessment were imputed using data available from the baseline assessment. The mean age difference between imaging assessment and the initial visit was 8.53 years (SD = 1.56 years). Sample sizes and descriptions for all the behavioural phenotypes can be found in Supplementary Data 1.

**Neuroimaging phenotypes.** Neuroimaging data consisted of: (1) intracranial and subcortical volumes ($N_{discovery}$ = 10,631 and $N_{replication}$ = 5553), containing eight major structures[46]; (2) T2 flair imaging for the whole brain ($N_{discovery}$ = 9829 and $N_{replication}$ = 5472) and in subcortical regions ($N_{discovery}$ = 9702 and $N_{replication}$ = 5187), which assess plausible white matter hyperintensity, (3) white matter microstructure, indexed by FA, MD, neurite density (ICVF), isotropic volume fraction and orientation dispersion index ($N_{discovery}$ = 9377 and $N_{replication}$ = 5239) for measures of white matter microstructure, in which we included three measures of association, projection and thalamic radiation subsets, and 15 major individual white matter tracts[26]; (4) pair-wise resting-state functional (rsfMRI) connectivity ($N_{discovery}$ = 9745 and $N_{replication}$ = 5241) of 21 nodes over the whole brain[28]; and finally (5) the amplitude of low-frequency rsfMRI signal fluctuation of the 21 nodes ($N_{discovery}$ = 9745 and $N_{replication}$ = 5241). All four types of neuroimaging data consisted of the imaging-derived phenotypes provided by UK Biobank. Available data for the Hariri faces/shapes emotion task included only whole-brain activation measures and a single region of interest (amygdala). We decided to exclude this sparse data from our analyses until more comprehensive measures become available. Images were acquired, pre-processed and quality controlled by UK Biobank using the FMRIB Software Library packages by a standard protocol (URL: https://biobank.ctsu.ox.ac.uk/crystal/docs/brain_mri.pdf), which was also described in two protocol papers[18,48]. All pilot study data with inconsistent scanner settings and data that did not pass the initial quality assessment conducted by UK Biobank imaging team were not included in the analysis. All imaging data were collected using a 3-T Siemens Skyra (software platform VD13) machine. For clarity, major steps of pre-processing are described in Supplementary Methods.

**Statistical models for PheWAS.** The GLM function in R (R version 3.2.3 and version 3.3.2, RStudio version 0.98.1080) was used to test the PheWAS associations[49], and the LME function from the 'nlme' package (version 3.1.131 under R version 3.2.3) in R[50] was used to test bilateral brain structures where hemisphere was included as a within-subject variable. Depression-PRSs were set as independent fixed effects, and behavioural and neuroimaging phenotypes were set as dependent variables. Overall, 552 phenotypes (209 behavioural phenotypes + 8 white matter hyperintensity measures + 9 intracranial/subcortical volumes + 95 diffusion tensor imaging measures + 210 rsfMRI connectivity + 21 rsfMRI fluctuation amplitude) × 8 depression-PRS (under 8 $p$ thresholds) = 4416 tests across phenotypes and depression-PRS $p$ thresholds were corrected altogether by FDR-correction[51] using p.adjust function in R ($q < 0.05$).

Covariates included in all association tests were sex, age, age², the first 15 genetic principal components and genotyping array[25]. For the replication analysis, MRI site was added in addition to the above covariates for all association tests. In addition to these covariates, adjustments were made for other confounders that were relevant to each phenotypic category, as listed below. Scanner positions on the $x$, $y$ and $z$ axes were included in the models for all brain phenotypes to control for static-field heterogeneity[38]. Mean head motion was set as a covariate for the rsfMRI data[28,52]. Subcortical volumetric tests controlled for intracranial volume[26,53]. Hemisphere was controlled for where applicable in bilateral brain structural phenotypes[26]. A list of covariates for each type of phenotype can be found in Supplementary Table 1. Distributions of PRS using different covariates can be found in Supplementary Fig. 17.

In order to help compare the results of logistic and linear regression models, we report the standardised regression coefficients for the models as effect sizes ($\beta$) for both types of models. Log-transformed odds ratio for binary dependent variables using logistic regression models are therefore reported. FDR-corrected $p$ values are reported throughout. For clarity, we have also reported the number of associations found using Bonferroni correction as a supplementary method. We acknowledge that as the phenotypes are likely to be correlated, and therefore Bonferroni correction is considered overly conservative. When effect sizes of different signs were presented together, we reported the range of absolute effect sizes. Two-side statistical tests were applied in all analyses.

**Replication analysis for PheWAS.** Traits that were found to be significantly associated with depression-PRS at a minimum of four PRS variant $p$ thresholds were selected for re-analysis in the independent replication sample. Our decision to combine FDR correction and four-threshold criterion was to control for type I error in the first step (achieved by FDR correction) and to carry the most robust and stable findings significant in more than half of all PRS thresholds to the

following MR and mediation analyses. This rationale is in line with studies on depression itself (depression-PRS predicting depression), whereby similar odds ratios are typically reported across multiple PRS thresholds[5,6]. The replication analysis was conducted on the selected traits across all eight depression-PRS thresholds. Results were considered to be replicated where they showed an identical direction of effect across discovery and replication samples and where the $p$ value for the replication sample analysis was significant after correction for multiple testing for depression-PRS at a minimum of four $p$ thresholds. FDR correction was applied to all the tests conducted in the replication analysis across all traits and p thresholds (e.g., if m traits were taken into replication analysis, then $p$ value adjustment was applied to all $m \times 8$ thresholds). The number of associations found using Bonferroni correction was also reported for clarity.

**Bidirectional MR analyses on depression and neuroimaging variables.** We used the 'twosampleMR' package version 0.4.22 in R to conduct bidirectional two-sample MR analyses between depression and neuroimaging variables in order to test for causal effects[54]. MR uses genetic data as instruments for testing whether there is any causal effect between an exposure and an outcome variable. A chart illustrating the underlying models can be found in Fig. 5, a flow chart of all the steps in Supplementary Fig. 18 and the main procedure is summarised below.

GWAS summary statistics for depression came from the meta-analysis used to generate the PRS as described above. For the neuroimaging variables, the ones that were found associated with depression-PRS in both the discovery and replication samples were chosen. GWAS were conducted using BGENIE version 3[55] on these neuroimaging variables in the UK Biobank imaging sample that were used in the PheWAS. Therefore, all exclusion criteria, genetic data quality check, ancestry control, relatedness removal and covariates remain the same as the depression GWAS. Overlapping individuals between the depression GWAS and the neuroimaging GWAS were removed. The neuroimaging variables were scaled to Mean = 0, SD = 1 to obtain standardised estimates. SNP heritability of depression and number of genome-wide significant hits are reported elsewhere[6]. SNP heritability of white matter microstructure measures estimated using linkage disequilibrium (LD) score regression[56] ranged from 13.2% to 34.0% and resting-state fluctuation amplitude ranged from 13.8% to 14.5%. The number of genome-wide significant loci ($p < 5 \times 10^{-8}$) ranged from 2 to 14 for all neuroimaging phenotypes. More details of neuroimaging GWAS summary statistics can be found in Supplementary Table 4.

To test the causal effect of depression on neuroimaging variables, genetic instruments were chosen from the GWAS summary statistics of depression[6], at a $p$ threshold of $5 \times 10^{-8}$. These SNPs were then clumped with a distance of 3000 kb and a maximum LD $r^2$ of 0.001, resulting in 107 independent genetic instruments. These SNPs were then identified within the GWAS summary statistics for each outcome, and those that were not present in both GWAS data sets were removed. SNP effect data on both the exposure and outcome were then harmonised to match the effect alleles before conducting the MR analyses.

For the causal effects of neuroimaging variables on depression, genetic instruments were chosen at a $p$ threshold of $8 \times 10^{-6}$, as the smallest number of genome-wide significant hits for neuroimaging GWAS was 2 prior to harmonising the two GWAS summary statistics. We therefore chose this lower threshold for neuroimaging GWAS to select genetic instruments. The same approach has been used in a previous MR study[57]. Genome-wide significant SNPs for depression and relevant genes have been reported and discussed by Howard et al.[6]. For significant MR results showing effects from neuroimaging measures to depression, we conducted manual inspections on scatter plots to ensure that the top neuroimaging GWAS SNPs driving the results were indeed brain relevant by checking if they have been associated gene expression in neural tissues or associated with other psychiatric or brain phenotypes in previous studies. SNPs that appear anomalous are reported in "Results" and highlighted in "Discussion". Genetic instruments used in the bidirectional MR analyses are reported in Supplementary Data 7 and Supplementary Table 3. At this threshold, after clumping with the same parameters as for choosing genetic instruments for depression, 12–44 independent genetic instruments were identified for each neuroimaging variable (see Supplementary Table 4). To further illustrate overlapping genetic architecture, we reported results for LD score regression based on the summary statistics above.

Three robust MR methods were chosen: MR-Egger, IVW, and the weighted median method. We also conducted three additional analyses (i) to test for horizontal pleiotropy by estimating the MR-Egger intercept and to test global heterogeneity of the genetic instruments using (ii) the Q test[54] and (iii) the MR-Presso global test (using R package 'MRPRESSO' version 1.0)[54]. Four types of plots were generated for visual inspection: (1) leave-one-out plot for testing SNP outliers, (2) funnel plot to show horizontal pleiotropy, (3) forest plot showing single SNP effects in the MR analysis, and finally, (4) scatter plot for overall inspection of effect sizes in GWAS for the cause and outcome.

FDR corrections were applied separately on each MR method within each trait category using a traditional whole-brain family-wise error correction as can be widely seen in other neuroimaging studies[59,60].

We have also provided results for genetic correlation using Linkage Disequilibrium Score Regression v1.0.0[56] in the main text and phenotypic association between depressive symptoms and other variables in Supplementary Table 4 and Supplementary Figs. 12–14 for completeness.

**Statistical models to test for the mediating effect of neuroimaging variables**. Following the PheWAS and MR analyses, we sought to test whether manifestations of depression were mediating the causal effect of depression-PRS on brain imaging phenotypes, as well as whether the neuroimaging variables act as neural mediators of genetic risk on depressive traits (i.e. neuroimaging traits were 'endophenotypes'). These tests were applied using SEM with the 'lavaan' package version 0.5.23.1097 in R v3.2.3[61]. Two types of mediation analysis were conducted (Supplementary Fig. 19). The first one aimed to test whether the neuroimaging effects were the consequence of depression by testing whether depression mediated the relationship between polygenic risk and neuroimaging variables (predictor = depression-PRS, mediator variable = CIDI definition of depression/depressive symptoms and dependent variables = neuroimaging traits). Neuroimaging variables were chosen from those measures that showed a significant causal effect from depression in the MR analyses. The second type of mediation models tested whether neuroimaging variables mediated the relationship between polygenic risk of depression on depressive phenotypes (predictor = depression-PRS, mediator = neuroimaging traits and dependent variable = CIDI definition of depression/depressive symptoms). The list of mediators was restricted to the neuroimaging phenotypes that showed significant causal effects on depression by MR analyses. For both types of mediation analyses, variables for manifestations of depression include CIDI definition for depression, severity of depression assessed by CIDI short form[62] and the current symptoms at the imaging assessment measured by PHQ-4[63]. In order to maximise statistic power, all mediation tests used the full sample that included both discovery and replication data sets ($N = 21,888$), adjusted for site.

All covariates remained the same as for PheWAS regression models. $p$ Value correction followed the same method as the MR analysis. Illustration for the models can be found in Supplementary Fig. 1, Supplementary Data 8 and Supplementary Methods.

**Interactions of depression-PRS and early risk factors or sociodemographic variables**. Interactions between environmental variables, previously associated with depression, and depression-PRS were tested. Environmental variables were chosen from early-life risk factors and sociodemographic variables previously found associated with risk for depression and showed depression case–control difference in the present sample ($p < 0.05$), which include: household income, Townsend Index, childhood trauma, adulthood trauma, and recent stressful life events in the past 6 months before imaging assessment[64,65]. Additional tests on the interaction effect between depression-PRS and sex were also reported in Supplementary Data 14 for completeness.

Dependent variables were the behavioural and imaging phenotypes that had significant associations with depression-PRS at a minimum of four thresholds in both the discovery and replication samples. Variables that were selected as factors were not included as dependent variables. The covariates included in these G × E analyses were those included in the PheWAS analyses, plus the interaction terms for PRS × covariates and environmental variables × covariates, in accordance with previous studies[66]. FDR correction was applied in the same manner with the PheWAS ($m$ dependent variables × 8 $p$ thresholds).

**Reporting summary**. Further information on research design is available in the Nature Research Reporting Summary linked to this article.

## Data availability
The data used in the present study is available from UK Biobank with restrictions applied. Data were used under license and thus not publicly available. Access to the UK Biobank data can be requested through a standard protocol (https://www.ukbiobank.ac.uk/register-apply/). The summary statistics of PGC_139k can be accessed from https://doi.org/10.7488/ds/2458. A data transfer agreement is required for accessing the 23andMe_307k summary statistics for the GWAS of depression (https://research.23andme.com/dataset-access/).

## Code availability
All code used for data preparation and analysis are available upon request.

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

## Acknowledgements

The present study is supported by a Wellcome Trust Strategic Award "Stratifying Resilience and Depression Longitudinally" (STRADL) (Reference 104036/Z/14/Z) and MRC Mental Health Data Pathfinder Award (Reference MC_PC_17209). Data acquisition and analyses were conducted using the UK Biobank Resource under approved project #4844. Participation of the UK Biobank subjects is gratefully appreciated. We also thank UK Biobank team for collecting and preparing data for analyses. Funding from the Biotechnology and Biological Sciences Research Council (BBSRC) and Medical Research Council (MRC) is gratefully acknowledged. The PGC has received major funding from the US National Institute of Mental Health (5 U01MH109528-03). We thank the research participants and employees of 23andMe for supporting this study. X.S. receives support from China Scholarship Council (No. 201506040037). H.C.W. is supported by a JMAS SIM fellowship from the Royal College of Physicians of Edinburgh and by an ESAT College Fellowship from the University of Edinburgh. D.M.H. is supported by a Sir Henry Wellcome Postdoctoral Fellowship (Reference 213674/Z/18/Z) and a 2018 NARSAD Young Investigator Grant from the Brain & Behavior Research Foundation (Reference 27404). W.D.H. is supported by a grant from Age UK (Disconnected Mind Project). A.M.M. is supported by the Sackler Trust. I.J.D. is a participant in UK Biobank.

## Author contributions

X.S., A.M.M. and H.C.W. contributed to the design of the study and writing the manuscript. X.S. conducted data analysis and data illustration. M.J.A. applied quality control for the genetic data from UK Biobank. D.M.H. and M.J.A. provided the protocol and scripts for all GWAS described in the manuscript. 23andMe Research Team and Major Depressive Disorder Working Group of the Psychiatric Genomics Consortium provided the summary statistics for depression GWAS conducted within their respective cohorts. W.D.H., T.-K.C. and I.J.D. provided expertise of study methodology and data presentation. A.M.M. and H.C.W. oversaw the project. All authors discussed and commented on the manuscript.

## Competing interests

The authors declare no competing interests.

## Additional information

## Major Depressive Disorder Working Group of the Psychiatric Genomics Consortium

Mark J. Adams[1], Toni-Kim Clarke[1], Andrew M. McIntosh[1,3,4], Ian J. Deary[3,4], Naomi R. Wray[5,6], Stephan Ripke[7,8,9], Manuel Mattheisen[10,11,12], Maciej Trzaskowski[5], Enda M. Byrne[5], Abdel Abdellaoui[13], Esben Agerbo[14,15,16], Tracy M. Air[17], Till F. M. Andlauer[18,19], Silviu-Alin Bacanu[20], Marie Bækvad-Hansen[16,21], Aartjan T. F. Beekman[22], Tim B. Bigdeli[20,23], Elisabeth B. Binder[18,24], Julien Bryois[25], Henriette N. Buttenschøn[16,26,27], Jonas Bybjerg-Grauholm[16,21], Na Cai[28,29], Enrique Castelao[30], Jane Hvarregaard Christensen[12,16,27], Jonathan R. I. Coleman[2], Lucía Colodro-Conde[31], Baptiste Couvy-Duchesne[6,32], Nick Craddock[33], Gregory E. Crawford[34,35], Gail Davies[3], Franziska Degenhardt[36], Eske M. Derks[31], Nese Direk[37,38], Conor V. Dolan[13], Erin C. Dunn[39,40,41], Thalia C. Eley[2], Valentina Escott-Price[42], Farnush Farhadi Hassan Kiadeh[43], Hilary K. Finucane[44,45], Jerome C. Foo[46], Andreas J. Forstner[36,47,48,49], Josef Frank[46], Héléna A. Gaspar[2], Michael Gill[50], Fernando S. Goes[51], Scott D. Gordon[31], Jakob Grove[12,16,27,52], Lynsey S. Hall[1,53], Christine Søholm Hansen[16,21], Thomas F. Hansen[54,55,56], Stefan Herms[36,48], Ian B. Hickie[57], Per Hoffmann[36,48], Georg Homuth[58], Carsten Horn[59], Jouke-Jan Hottenga[13], David M. Hougaard[13,18], David M. Howard[1,2], Marcus Ising[60], Rick Jansen[22], Ian Jones[61], Lisa A. Jones[62], Eric Jorgenson[63], James A. Knowles[64], Isaac S. Kohane[65,66,67], Julia Kraft[8], Warren W. Kretzschmar[68], Zoltán Kutalik[69,70], Yihan Li[68], Penelope A. Lind[31], Donald J. MacIntyre[71,72], Dean F. MacKinnon[51], Robert M. Maier[6], Wolfgang Maier[73], Jonathan Marchini[74], Hamdi Mbarek[13], Patrick McGrath[75], Peter McGuffin[2], Sarah E. Medland[31], Divya Mehta[6,76], Christel M. Middeldorp[13,77,78], Evelin Mihailov[79], Yuri Milaneschi[22], Lili Milani[79], Francis M. Mondimore[51], Grant W. Montgomery[5], Sara Mostafavi[80,81], Niamh Mullins[2], Matthias Nauck[82,83], Bernard Ng[81], Michel G. Nivard[13], Dale R. Nyholt[84], Paul F. O'Reilly[2], Hogni Oskarsson[85], Michael J. Owen[61], Jodie N. Painter[31], Carsten Bøcker Pedersen[14,15,16], Marianne Giørtz Pedersen[14,15,16], Roseann E. Peterson[20,86], Erik Pettersson[25], Wouter J. Peyrot[22], Giorgio Pistis[30], Danielle Posthuma[87,88], Jorge A. Quiroz[89], Per Qvist[12,16,27], John P. Rice[90], Brien P. Riley[20], Margarita Rivera[2,91], Saira Saeed Mirza[37], Robert Schoevers[92], Eva C. Schulte[93,94], Ling Shen[63], Jianxin Shi[95], Stanley I. Shyn[96], Engilbert Sigurdsson[97], Grant C. B. Sinnamon[98], Johannes H. Smit[22], Daniel J. Smith[99], Hreinn Stefansson[100], Stacy Steinberg[100], Fabian Streit[46], Jana Strohmaier[46], Katherine E. Tansey[101], Henning Teismann[102], Alexander Teumer[103], Wesley Thompson[16,55,104,105], Pippa A. Thomson[106], Thorgeir E. Thorgeirsson[100], Matthew Traylor[107], Jens Treutlein[46], Vassily Trubetskoy[8], Andrés G. Uitterlinden[108], Daniel Umbricht[109], Sandra Van der Auwera[110], Albert M. van Hemert[111], Alexander Viktorin[25], Peter M. Visscher[5,6], Yunpeng Wang[16,55,105], Bradley T. Webb[112], Shantel Marie Weinsheimer[16,55], Jürgen Wellmann[102], Gonneke Willemsen[13], Stephanie H. Witt[46], Yang Wu[5], Hualin S. Xi[113], Jian Yang[6,114], Futao Zhang[5], Volker Arolt[115], Bernhard T. Baune[116,117,118], Klaus Berger[102], Dorret I. Boomsma[13], Sven Cichon[36,48,119,120], Udo Dannlowski[115], E. J. C. de Geus[13,121], J. Raymond DePaulo[51], Enrico Domenici[122], Katharina Domschke[123,124], Tõnu Esko[9,79], Hans J. Grabe[110], Steven P. Hamilton[125], Caroline Hayward[126], Andrew C. Heath[90], Kenneth S. Kendler[20], Stefan Kloiber[60,127,128], Glyn Lewis[129], Qingqin S. Li[130], Susanne Lucae[60], Pamela A. F. Madden[90], Patrik K. Magnusson[25], Nicholas G. Martin[31], Andres Metspalu[79,131], Ole Mors[16,132], Preben Bo Mortensen[14,15,16,27], Bertram Müller-Myhsok[18,133,134], Merete Nordentoft[16,135], Markus M. Nöthen[36], Michael C. O'Donovan[61], Sara A. Paciga[136], Nancy L. Pedersen[25], Brenda W. J. H. Penninx[22], Roy H. Perlis[39,137], David J. Porteous[106], James B. Potash[138], Martin Preisig[30],

Marcella Rietschel[46], Catherine Schaefer[63], Thomas G. Schulze[46,94,139,140,141], Jordan W. Smoller[39,40,41], Kari Stefansson[100,142], Henning Tiemeier[37,143,144], Rudolf Uher[145], Henry Völzke[103], Myrna M. Weissman[75,146], Thomas Werge[16,55,147], Cathryn M. Lewis[2,148], Douglas F. Levinson[149], Gerome Breen[2,150], Anders D. Børglum[12,16,27] & Patrick F. Sullivan[25,151,152]

[5]Institute for Molecular Bioscience, The University of Queensland, Brisbane, QLD, Australia. [6]Queensland Brain Institute, The University of Queensland, Brisbane, QLD, Australia. [7]Analytic and Translational Genetics Unit, Massachusetts General Hospital, Boston, MA, USA. [8]Department of Psychiatry and Psychotherapy, Universitätsmedizin Berlin Campus Charité Mitte, Berlin, DE, Germany. [9]Medical and Population Genetics, Broad Institute, Cambridge, MA, USA. [10]Department of Psychiatry, Psychosomatics and Psychotherapy, University of Wurzburg, Wurzburg, DE, Germany. [11]Centre for Psychiatry Research, Department of Clinical Neuroscience, Karolinska Institutet, Stockholm, Sweden. [12]Department of Biomedicine, Aarhus University, Aarhus, Denmark. [13]Dept of Biological Psychology & EMGO+Institute for Health and Care Research, Vrije Universiteit Amsterdam, Amsterdam, Netherlands. [14]Centre for Integrated Register-based Research, Aarhus University, Aarhus, Denmark. [15]National Centre for Register-Based Research, Aarhus University, Aarhus, Denmark. [16]iPSYCH, The Lundbeck Foundation Initiative for Integrative Psychiatric Research, Copenhagen, Denmark. [17]Discipline of Psychiatry, University of Adelaide, Adelaide, SA, Australia. [18]Department of Translational Research in Psychiatry, Max Planck Institute of Psychiatry, Munich, Germany. [19]Department of Neurology, Klinikum rechts der Isar, Technical University of Munich, Munich, Germany. [20]Department of Psychiatry, Virginia Commonwealth University, Richmond, VA, USA. [21]Center for Neonatal Screening, Department for Congenital Disorders, Statens Serum Institut, Copenhagen, Denmark. [22]Department of Psychiatry, Vrije Universiteit Medical Center and GGZ inGeest, Amsterdam, Netherlands. [23]Virginia Institute for Psychiatric and Behavior Genetics, Richmond, VA, USA. [24]Department of Psychiatry and Behavioral Sciences, Emory University School of Medicine, Atlanta, GA, USA. [25]Department of Medical Epidemiology and Biostatistics, Karolinska Institutet, Stockholm, Sweden. [26]Department of Clinical Medicine, Translational Neuropsychiatry Unit, Aarhus University, Aarhus, Denmark. [27]iSEQ, Centre for Integrative Sequencing, Aarhus University, Aarhus, Denmark. [28]Human Genetics, Wellcome Trust Sanger Institute, Cambridge, Great Britain. [29]Statistical genomics and systems genetics, European Bioinformatics Institute (EMBL-EBI), Cambridge, Great Britain. [30]Department of Psychiatry, University Hospital of Lausanne, Prilly, Vaud, Switzerland. [31]Genetics and Computational Biology, QIMR Berghofer Medical Research Institute, Brisbane, QLD, Australia. [32]Centre for Advanced Imaging, The University of Queensland, Brisbane, QLD, Australia. [33]Psychological Medicine, Cardiff University, Cardiff, Great Britain. [34]Center for Genomic and Computational Biology, Duke University, Durham, NC, USA. [35]Department of Pediatrics, Division of Medical Genetics, Duke University, Durham, NC, USA. [36]Institute of Human Genetics, University of Bonn, School of Medicine & University Hospital Bonn, Bonn, Germany. [37]Epidemiology, Erasmus MC, Rotterdam, Zuid-Holland, Netherlands. [38]Psychiatry, Dokuz Eylul University School Of Medicine, Izmir, Turkey. [39]Department of Psychiatry, Massachusetts General Hospital, Boston, MA, USA. [40]Psychiatric and Neurodevelopmental Genetics Unit (PNGU), Massachusetts General Hospital, Boston, MA, USA. [41]Stanley Center for Psychiatric Research, Broad Institute, Cambridge, MA, USA. [42]Neuroscience and Mental Health, Cardiff University, Cardiff, Great Britain. [43]Bioinformatics, University of British Columbia, Vancouver, BC, Canada. [44]Department of Epidemiology, Harvard T.H. Chan School of Public Health, Boston, MA, USA. [45]Department of Mathematics, Massachusetts Institute of Technology, Cambridge, MA, USA. [46]Department of Genetic Epidemiology in Psychiatry, Central Institute of Mental Health, Medical Faculty Mannheim, Heidelberg University, Mannheim, Baden-Württemberg, Germany. [47]Department of Psychiatry (UPK), University of Basel, Basel, Switzerland. [48]Department of Biomedicine, University of Basel, Basel, Switzerland. [49]Centre for Human Genetics, University of Marburg, Marburg, Germany. [50]Department of Psychiatry, Trinity College Dublin, Dublin, Ireland. [51]Psychiatry & Behavioral Sciences, Johns Hopkins University, Baltimore, MD, USA. [52]Bioinformatics Research Centre, Aarhus University, Aarhus, Denmark. [53]Institute of Genetic Medicine, Newcastle University, Newcastle upon Tyne, Great Britain. [54]Danish Headache Centre, Department of Neurology, Rigshospitalet, Glostrup, Denmark. [55]Institute of Biological Psychiatry, Mental Health Center Sct. Hans, Mental Health Services Capital Region of Denmark, Copenhagen, Denmark. [56]iPSYCH, The Lundbeck Foundation Initiative for Psychiatric Research, Copenhagen, Denmark. [57]Brain and Mind Centre, University of Sydney, Sydney, NSW, Australia. [58]Interfaculty Institute for Genetics and Functional Genomics, Department of Functional Genomics, University Medicine and Ernst Moritz Arndt University Greifswald, Greifswald, Mecklenburg-Vorpommern, Germany. [59]Roche Pharmaceutical Research and Early Development, Pharmaceutical Sciences, Roche Innovation Center Basel, F. Hoffmann-La Roche Ltd, Basel, Switzerland. [60]Max Planck Institute of Psychiatry, Munich, Germany. [61]MRC Centre for Neuropsychiatric Genetics and Genomics, Cardiff University, Cardiff, Great Britain. [62]Department of Psychological Medicine, University of Worcester, Worcester, Great Britain. [63]Division of Research, Kaiser Permanente Northern California, Oakland, CA, USA. [64]Psychiatry & The Behavioral Sciences, University of Southern California, Los Angeles, CA, USA. [65]Department of Biomedical Informatics, Harvard Medical School, Boston, MA, USA. [66]Department of Medicine, Brigham and Women's Hospital, Boston, MA, USA. [67]Informatics Program, Boston Children's Hospital, Boston, MA, USA. [68]Wellcome Trust Centre for Human Genetics, University of Oxford, Oxford, Great Britain. [69]Institute of Social and Preventive Medicine (IUMSP), University Hospital of Lausanne, Lausanne, VD, Switzerland. [70]Swiss Institute of Bioinformatics, Lausanne, VD, Switzerland. [71]Division of Psychiatry, Centre for Clinical Brain Sciences, University of Edinburgh, Edinburgh, Great Britain. [72]Mental Health, NHS 24, Glasgow, Great Britain. [73]Department of Psychiatry and Psychotherapy, University of Bonn, Bonn, Germany. [74]Statistics, University of Oxford, Oxford, Great Britain. [75]Psychiatry, Columbia University College of Physicians and Surgeons, New York, NY, USA. [76]School of Psychology and Counseling, Queensland University of Technology, Brisbane, QLD, Australia. [77]Child and Youth Mental Health Service, Children's Health Queensland Hospital and Health Service, South Brisbane, QLD, Australia. [78]Child Health Research Centre, University of Queensland, Brisbane, QLD, Australia. [79]Estonian Genome Center, University of Tartu, Tartu, Estonia. [80]Medical Genetics, University of British Columbia, Vancouver, BC, Canada. [81]Statistics, University of British Columbia, Vancouver, BC, Canada. [82]DZHK (German Centre for Cardiovascular Research), Partner Site Greifswald, University Medicine, University Medicine Greifswald, Greifswald, Mecklenburg-Vorpommern, Germany. [83]Institute of Clinical Chemistry and Laboratory Medicine, University Medicine Greifswald, Greifswald, Mecklenburg-Vorpommern, Germany. [84]Institute of Health and Biomedical Innovation, Queensland University of Technology, Brisbane, QLD, Australia. [85]Humus, Reykjavik, Iceland. [86]Virginia Institute for Psychiatric & Behavioral Genetics, Virginia Commonwealth University, Richmond, VA, USA. [87]Clinical Genetics, Vrije Universiteit Medical Center, Amsterdam, Netherlands. [88]Complex Trait Genetics, Vrije Universiteit Amsterdam, Amsterdam, Netherlands. [89]Solid Biosciences, Boston, MA, USA. [90]Department of Psychiatry, Washington University in Saint Louis School of Medicine, Saint Louis, MO, USA. [91]Department of Biochemistry and Molecular Biology II, Institute of Neurosciences, Center for Biomedical Research, University of Granada, Granada, Spain. [92]Department of Psychiatry, University of Groningen, University Medical Center Groningen, Groningen, Netherlands. [93]Department of Psychiatry and Psychotherapy, University Hospital, Ludwig Maximilian University Munich, Munich, Germany. [94]Institute of Psychiatric Phenomics and Genomics (IPPG), University Hospital, Ludwig Maximilian University Munich, Munich, Germany. [95]Division of Cancer Epidemiology and Genetics, National

Cancer Institute, Bethesda, MD, USA. [96]Behavioral Health Services, Kaiser Permanente Washington, Seattle, WA, USA. [97]Faculty of Medicine, Department of Psychiatry, University of Iceland, Reykjavik, Iceland. [98]School of Medicine and Dentistry, James Cook University, Townsville, QLD, Australia. [99]Institute of Health and Wellbeing, University of Glasgow, Glasgow, Great Britain. [100]deCODE Genetics/Amgen, Reykjavik, Iceland. [101]College of Biomedical and Life Sciences, Cardiff University, Cardiff, Great Britain. [102]Institute of Epidemiology and Social Medicine, University of Münster, Münster, Nordrhein-Westfalen, Germany. [103]Institute for Community Medicine, University Medicine Greifswald, Greifswald, Mecklenburg-Vorpommern, Germany. [104]Department of Psychiatry, University of California, San Diego, San Diego, CA, USA. [105]KG Jebsen Centre for Psychosis Research, Norway Division of Mental Health and Addiction, Oslo University Hospital, Oslo, Norway. [106]Medical Genetics Section, CGEM, IGMM, University of Edinburgh, Edinburgh, Great Britain. [107]Clinical Neurosciences, University of Cambridge, Cambridge, Great Britain. [108]Internal Medicine, Erasmus MC, Rotterdam, Zuid-Holland, Netherlands. [109]Roche Pharmaceutical Research and Early Development, Neuroscience, Ophthalmology and Rare Diseases Discovery & Translational Medicine Area, Roche Innovation Center Basel, F. Hoffmann-La Roche Ltd, Basel, Switzerland. [110]Department of Psychiatry and Psychotherapy, University Medicine Greifswald, Greifswald, Mecklenburg-Vorpommern, Germany. [111]Department of Psychiatry, Leiden University Medical Center, Leiden, Netherlands. [112]Virginia Institute for Psychiatric & Behavioral Genetics, Virginia Commonwealth University, Richmond, VA, USA. [113]Computational Sciences Center of Emphasis, Pfizer Global Research and Development, Cambridge, MA, USA. [114]Institute for Molecular Bioscience; Queensland Brain Institute, The University of Queensland, Brisbane, QLD, Australia. [115]Department of Psychiatry, University of Münster, Münster, Nordrhein-Westfalen, Germany. [116]Department of Psychiatry, University of Münster, Münster, Germany. [117]Department of Psychiatry, Melbourne Medical School, University of Melbourne, Melbourne, VIC, Australia. [118]Florey Institute for Neuroscience and Mental Health, University of Melbourne, Melbourne, VIC, Australia. [119]Institute of Medical Genetics and Pathology, University Hospital Basel, University of Basel, Basel, Switzerland. [120]Institute of Neuroscience and Medicine (INM-1), Research Center Juelich, Juelich, Germany. [121]Amsterdam Public Health Institute, Vrije Universiteit Medical Center, Amsterdam, Netherlands. [122]Centre for Integrative Biology, Università degli Studi di Trento, Trento, Trentino-Alto Adige, Italy. [123]Department of Psychiatry and Psychotherapy, Medical Center - University of Freiburg, Faculty of Medicine, University of Freiburg, Freiburg, Germany. [124]Center for NeuroModulation, Faculty of Medicine, University of Freiburg, Freiburg, Germany. [125]Psychiatry, Kaiser Permanente Northern California, San Francisco, CA, USA. [126]Medical Research Council Human Genetics Unit, Institute of Genetics and Molecular Medicine, University of Edinburgh, Edinburgh, Great Britain. [127]Department of Psychiatry, University of Toronto, Toronto, ON, Canada. [128]Centre for Addiction and Mental Health, Toronto, ON, Canada. [129]Division of Psychiatry, University College London, London, Great Britain. [130]Neuroscience Therapeutic Area, Janssen Research and Development, LLC, Titusville, NJ, USA. [131]Institute of Molecular and Cell Biology, University of Tartu, Tartu, Estonia. [132]Psychosis Research Unit, Aarhus University Hospital, Risskov, Denmark. [133]Munich Cluster for Systems Neurology (SyNergy), Munich, Germany. [134]University of Liverpool, Liverpool, Great Britain. [135]Mental Health Center Copenhagen, Copenhagen Universtity Hospital, Copenhagen, Denmark. [136]Human Genetics and Computational Biomedicine, Pfizer Global Research and Development, Groton, CT, USA. [137]Psychiatry, Harvard Medical School, Boston, MA, USA. [138]Psychiatry, University of Iowa, Iowa City, IA, USA. [139]Department of Psychiatry and Behavioral Sciences, Johns Hopkins University, Baltimore, MD, USA. [140]Department of Psychiatry and Psychotherapy, University Medical Center Göttingen, Goettingen, Niedersachsen, Germany. [141]Human Genetics Branch, NIMH Division of Intramural Research Programs, Bethesda, MD, USA. [142]Faculty of Medicine, University of Iceland, Reykjavik, Iceland. [143]Child and Adolescent Psychiatry, Erasmus MC, Rotterdam, Zuid-Holland, Netherlands. [144]Psychiatry, Erasmus MC, Rotterdam, Zuid-Holland, Netherlands. [145]Psychiatry, Dalhousie University, Halifax, NS, Canada. [146]Division of Epidemiology, New York State Psychiatric Institute, New York, NY, USA. [147]Department of Clinical Medicine, University of Copenhagen, Copenhagen, Denmark. [148]Department of Medical & Molecular Genetics, King's College London, London, Great Britain. [149]Psychiatry & Behavioral Sciences, Stanford University, Stanford, CA, USA. [150]NIHR Maudsley Biomedical Research Centre, King's College London, London, Great Britain. [151]Genetics, University of North Carolina at Chapel Hill, Chapel Hill, NC, USA. [152]Psychiatry, University of North Carolina at Chapel Hill, Chapel Hill, NC, USA.

