## [Peer Review File · Nature Communications]

Reviewers' Comments:

Reviewer #1:

Remarks to the Author:

Many aspects of this resubmission greatly improved this paper (e.g., including covariate x predictor interactions in the moderation models and clarifying methods). However, I still find it difficult to integrate this information into a broader picture that limits the utility of the manuscript in its present form. Presently, many findings are presented with the reader left to do a lot of integration.

1. Given the breadth and division of behavioral and neural phenotypes in this manuscript, a discussion of the broad pattern of findings would be useful. I would encourage the authors to explicitly discuss the utility of PRS derived from GWAS of psychiatric diagnosis with regard to the differential prediction between behavioral and neural phenotypes. Specifically, among the 209 behavioral phenotypes tested, 51 (i.e., 24.4%) showed replicable associations with PRS. Among the 343 neuroimaging phenotypes tested, only 26 (i.e., 7.6%) showed replicable association. Broadly, these results would seem to suggest that PRS derived from psychiatric diagnosis or behavioral phenotypes may not be as applicable to neural phenotypes. The authors are encouraged to directly discuss this broad pattern of findings. The following reference: PMID: 26854805, discusses the lack of additional power of neuroimaging phenotypes relative to psychiatric diagnoses for single variant association and may be useful. See also initial reviewer 1 comment 1AB – in addition to directly mentioning this the authors are encouraged to discuss reasons for this general pattern? For example, might SNP-based heritability differences or reliability concerns with rsfc of limited duration not make these ideal phenotypes for the study of individual differences, particularly for expected small effects of PRS (as opposed to brain – broad behavioral associations). A general broad discussion of the extent of association across behavioral vs neural phenotypes in the discussion would help move the field forward.

2. MR and mediation analyses. The authors very nicely and explicitly discuss the problems of comparing the MR models with depression vs neuroimaging genetic instruments due to the potential reduced instrumental validity of the latter due to less well powered GWAS and less significant loci being identified. Arguably the difference in power of the discovery GWAS alone makes comparing these different models extremely problematic – e.g., is the lack of brain causing depression a true negative or a false negative due to the underpowered initial GWAS – the authors now nicely discuss this. Further, I would also recommend that the authors consider their phenotypic data. Indeed, given that this sample contained older individuals and that depression typically onsets in adolescence – young/middle adulthood, it is possible that genomic risk for neural phenotypes may influence depression risk at younger ages, with then neural phenotypes assessed at an older age being more influenced by the expression of depression. I would encourage the authors to raise this possibility. Psychiatric diagnoses and their consequences play out in a complex dynamic and interaction developmental framework. Advancing conclusions based on one study that depression causes brain differences band that brain differences do not play an etiologic role in the expression of depression may overstep what these cross sectional data can truly tell us. See also initial reviewer 1 comment 2B.

Related, it would be useful to add a demographic table describing sample characteristics would be

useful and reporting age and gender in the main manuscript.

With regard to multiple testing correction, the authors note that correction was done within each trait category – does this mean within brain phenotypes and within behavioral phenotypes? Greater clarity would be useful here.

Given that we know PRS are typically most predictive of the same exact phenotype or the available phenotype that most closely approximates the phenotype in the discovery GWAS, will mediation analyses be inherently biased to finding support for the GWAS phenotype being a mediator: i.e., because this would inflate the A path with this as the mediator or the C path if it is the outcome. And if so, how can a comparison of these different models and arrangements inform understanding above and beyond the basic associations?

The mediational models are reported as documenting and testing causality. However, there is great controversy surrounding interpreting mediation models in this capacity which may be better conceptualized as indirect associations – e.g., <https://stats.stackexchange.com/questions/13340/are-mediation-analyses-inherently-causal>

Table S6A has many “NAs” listed for p.corrrected values – e.g., MD in ATR - why is this?

B. Given concerns about pleiotropy, the authors could include potentially confounding pleiotropic variables (or indicators of genetic risk) in the MR and mediational models to account for these factors – e.g., the other behavioral phenotypes associated. E.g., see sensitivity analyses accounting for genomic risk for tobacco in MR of schizophrenia and cannabis: PMID: 28115737 – however, since the pleiotropic phenotypes are actually measured in this study, they could go further and account for them in these analyses (e.g., including as covariates any genetic traits that show genetic correlations with depression).

C. a Relatively new technique is latent causal variable analysis approaches to MR, it is recommended that the authors add this as an MR analytic approach: PMID: 30374074. As this is a newer approach, applying this only to variables that made it through the other tests of MR would be reasonable.

3. Given that individuals were recruited from specific sites and scanned on different scanners, the data should be analyzed in a nested fashion using a mixed-effects model (e.g., lme4). However, it is unclear whether this was done based on the stat models for PheWas section.

4. The link to the HPA axis discussed on page 20 is unclear. One could make a similar argument for a host of different mechanistic systems. Greater rationalization is needed here for the authors to stipulate that the genetic correlations with depression suggest involvement of the HPA axis (page 20).

Reviewer #3:

Remarks to the Author:

The authors have done an excellent job in addressing my reviewer comments. I have no further comments.

- In the interest of accountability and transparency during the review process, I choose to sign my review, Lea K. Davis.

Point-to-point response

We would like to thank the editor and reviewers for their detailed considerations of our manuscript. We have now addressed reviewer 1's remaining concerns in a revised version.

Reviewer #1

Remarks to the Author:

Many aspects of this resubmission greatly improved this paper (e.g., including covariate x predictor interactions in the moderation models and clarifying methods). However, I still find it difficult to integrate this information into a broader picture that limits the utility of the manuscript in its present form. Presently, many findings are presented with the reader left to do a lot of integration.

We would like to thank the reviewers for their comments and for suggesting PMID: 26854805, which we now cite. We have now further summarised the findings and a single additional analysis has also been conducted and added to the manuscript addressing each point raised below (see point #1 below). We agree that this additional effort to integrate the findings of our analyses has improved the paper further.

1. Given the breadth and division of behavioral and neural phenotypes in this manuscript, a discussion of the broad pattern of findings would be useful. I would encourage the authors to explicitly discuss the utility of PRS derived from GWAS of psychiatric diagnosis with regard to the differential prediction between behavioral and neural phenotypes. Specifically, among the 209 behavioral phenotypes tested, 51 (i.e., 24.4%) showed replicable associations with PRS. Among the 343 neuroimaging phenotypes tested, only 26 (i.e., 7.6%) showed replicable association. Broadly, these results would seem to suggest that PRS derived from psychiatric diagnosis or behavioral phenotypes may not be as applicable to neural phenotypes. The authors are encouraged to directly discuss this broad pattern of findings. The following reference: PMID: 26854805, discusses the lack of additional power of neuroimaging phenotypes relative to psychiatric diagnoses for single variant association and may be useful. See also initial reviewer 1 comment 1AB – in addition to directly mentioning this the authors are encouraged to discuss reasons for this general pattern? For example, might SNP-based heritability differences or reliability concerns with rsfc of limited duration not make these ideal phenotypes for the study of individual differences, particularly for expected small effects of PRS (as opposed to brain – broad behavioral associations). A general broad discussion of the extent of association across behavioral vs neural phenotypes in the discussion would help move the field forward.

We have added discussion that explicitly address the comparisons between behavioural and neuroimaging variables on page 9, reads as,

“Whilst associations were found between depression PRS and both behavioural and neuroimaging variables, the proportion of replicated findings was relatively small for neuroimaging variables (7.6%), compared to behavioural phenotypes (24.4%). The high proportion found for behavioural phenotypes likely reflects the overlapping genetic architecture of multiple psychiatric conditions

with one another, and the behavioural traits with which they are commonly associated. In contrast, although several brain phenotypes were associated with depression PRS, our findings suggest that depression shares its genetic architecture with only a small proportion of them. One potential reason for the finding is a relative lack of signal in the GWAS of neuroimaging variables¹. It is also possible that depression may have a relatively specific relationship with a much smaller number of neuroimaging variables, reflecting the underlying mechanisms of depression.”

We have already addressed the short fMRI acquisition times, as a potential limitation of this study, elsewhere in our manuscript on page 14.

2. MR and mediation analyses. The authors very nicely and explicitly discuss the problems of comparing the MR models with depression vs neuroimaging genetic instruments due to the potential reduced instrumental validity of the latter due to less well powered GWAS and less significant loci being identified. Arguably the difference in power of the discovery GWAS alone makes comparing these different models extremely problematic – e.g., is the lack of brain causing depression a true negative or a false negative due to the underpowered initial GWAS – the authors now nicely discuss this. Further, I would also recommend that the authors consider their phenotypic data. Indeed, given that this sample contained older individuals and that depression typically onsets in adolescence – young/middle adulthood, it is possible that genomic risk for neural phenotypes may influence depression risk at younger ages, with then neural phenotypes assessed at an older age being more influenced by the expression of depression. I would encourage the authors to raise this possibility. Psychiatric diagnoses and their consequences play out in a complex dynamic and interaction developmental framework. Advancing conclusions based on one study that depression causes brain differences band that brain differences do not play an etiologic role in the expression of depression may overstep what these cross sectional data can truly tell us. See also initial reviewer 1 comment 2B.

We thank the reviewer for their comment and agree that the age of the participants in this study is relevant to its interpretation. In response to this issue, we have added the following text on page 13 in order to address this point:

“Participants in this study were in their mid- to later-life. Various factors, including ageing², the long-term effects of early developmental deficits^{3,4} and comorbid illnesses may impact variation in brain phenotypes in this age range. These possible explanations are difficult to disentangle in studies, such as UK Biobank, that currently lack longitudinal imaging data over a wide age-range. Studies of high-risk participants would also be useful, as they are able to make stronger causal inferences on the timing and trajectory of brain differences before and after the onset of illness.”

Related, it would be useful to add a demographic table describing sample characteristics would be useful and reporting age and gender in the main manuscript.

Age and sex were reported in the participants section in the main text. Age and sex for the discovery, replication and meta-analytic samples and other demographic information, such as education, household income and social deprivation, are added in Supplementary Table 17.

With regard to multiple testing correction, the authors note that correction was done within each trait category – does this mean within brain phenotypes and within behavioural phenotypes? Greater clarity would be useful here.

For PheWAS, multiple correction was applied for all 4,416 tests (on page 19). For MR, as a follow-

up analysis after PheWAS, multiple testing correction was applied within each neuroimaging category (e.g. FA/MD/ICVF/resting-state, details described on page 22). This has been highlighted in sections that describe PheWAS and MR analyses in the according paragraphs separately (pages 19 and 22).

Given that we know PRS are typically most predictive of the same exact phenotype or the available phenotype that most closely approximates the phenotype in the discovery GWAS, will mediation analyses be inherently biased to finding support for the GWAS phenotype being a mediator: i.e., because this would inflate the A path with this as the mediator or the C path if it is the outcome. And if so, how can a comparison of these different models and arrangements inform understanding above and beyond the basic associations?

The mediational models are reported as documenting and testing causality. However, there is great controversy surrounding interpreting mediation models in this capacity which may be better conceptualized as indirect associations – e.g., <https://stats.stackexchange.com/questions/13340/are-mediation-analyses-inherently-causal>

We agree with the reviewer that the strongest associations between depression-PRS are likely to be with depression and related symptoms. We also consider that mediation analysis alone may have its limitations, and for that reason we used this complementary method in addition to MR.

We thank the reviewer for the suggestion of changing the terminology from ‘mediation model’ to ‘indirect association’. We agree that the terminology used here is not uncontroversial, but we would prefer to retain the original wording approved by the previous reviewers, and in widespread use throughout the psychiatric genetics literature.

Table S6A has many “NAs” listed for p.corrected values – e.g., MD in ATR - why is this?

This was due to misalignment of the data. It is now changed into the correct format. This table is now Supplementary Table 10 due to reformatting.

B. Given concerns about pleiotropy, the authors could include potentially confounding pleiotropic variables (or indicators of genetic risk) in the MR and mediational models to account for these factors – e.g., the other behavioral phenotypes associated. E.g., see sensitivity analyses accounting for genomic risk for tobacco in MR of schizophrenia and cannabis: PMID: 28115737 – however, since the pleiotropic phenotypes are actually measured in this study, they could go further and account for them in these analyses (e.g., including as covariates any genetic traits that show genetic correlations with depression).

We appreciate the reviewer's suggestions for validating the MR results.

The first sensitivity analysis similar with PMID: 28115737 is the leave-one-out analysis. It has been reported in the supplementary materials (Supplementary Figures 7-11, Leave-one-out plots). For the most robust results shown in Figure 5, all leave-one-out tests remained significant, indicating that the results were not driven by a single genetic variant, but by a consistent pattern among all variants. These are now highlighted in the supplementary materials (page 13).

We also conducted another sensitivity analysis similar to PMID: 28115737 by removing possible pleiotropic genetic variants. We considered neuroticism as a major possible pleiotropic variable as

it showed the strongest genetic correlation with depression in Howard et al.'s (2019) GWAS paper⁵ (see Supplementary Table 3 in the reference paper). We removed the genome-wide significant SNPs in depression GWAS in medium to high LD with neuroticism ($r > 0.1$, four genetic instruments were thus removed), and conducted another MR to test the causal relationship between depression and brain, the results remained consistent with the initial findings. For the most robust findings reported in Figure 5, all remained significant in at least two MR methods (see Supplementary Table 20). Details are described in the supplementary materials, page 13.

We consider that including all 51 behavioural phenotypes in the GWAS is beyond the scope of this study. Another concern is that by adding all covariates we may introduce collider bias⁶. In the current form, we think the added sensitivity analysis considering the effect of neuroticism have increased our confidence in the results. Future efforts of discovering the relationship between the behavioural and neuroimaging variables are indeed important, and it is now discussed in the main text, page 11.

C. A relatively new technique is latent causal variable analysis approaches to MR, it is recommended that the authors add this as an MR analytic approach: PMID: 30374074. As this is a newer approach, applying this only to variables that made it through the other tests of MR would be reasonable.

We appreciate the reviewer for the suggestion. The present study contains a large volume of analysis including replication, four different MR analyses, various sensitivity analysis for MR findings, and mediation analysis, which have shown converging results. We consider this new method is beyond the scope of the present paper, and it is fundamentally different from MR analyses.

To use the new method as a confirmatory approach is concerning, because a lack of genetic correlation between depression and neuroimaging variables would lead to overly conservative p values, as suggested by the original paper proposing this method (page 1729)⁶. We also note that this latent component variance method takes the whole genome into account, whereas MR methods use the strongest genetic variants, which is more suitable for the summary statistics of GWAS for neuroimaging variables.

Therefore we think it is suitable to leave this analysis for future investigations when GWAS of neuroimaging traits with greater power become available.

3. Given that individuals were recruited from specific sites and scanned on different scanners, the data should be analyzed in a nested fashion using a mixed-effects model (e.g., lme4). However, it is unclear whether this was done based on the stat models for PheWas section.

Participants were recruited and data were primarily assessed with the imaging assessments, and therefore in our model we controlled for the site effect based on which site they were scanned at. For the online follow-up session, which has a unified form of computerised protocol, we think adding MRI site would be sufficient to control for site effect.

We replicated the PheWAS in the total sample (N=21,888), including assessment centre at recruitment as an additional covariate, the results remained very similar (r between effect sizes for

including/not including recruiting assessment centre > 0.99), and no originally significant findings turned null ($p_{\text{FDR}} < 0.017$, corrected across all significant associations found for the original model).

4. The link to the HPA axis discussed on page 20 is unclear. One could make a similar argument for a host of different mechanistic systems. Greater rationalization is needed here for the authors to stipulate that the genetic correlations with depression suggest involvement of the HPA axis (page 20).

We agree with the reviewer that a number of different mechanistic explanations are possible. We have now shortened this section, and we are no longer making strong inferences about this mechanism, which is not specifically tested in our analyses. The new discussion reads below (changes highlighted in blue on page 11 in the main text),

“The associations found between behavioural traits and depression-PRS suggest that polygenic risk of depression may also identify a predisposition to experience particular environmental risk exposures, or a vulnerability to their effects and later recall. Firstly, the linear association of depression-PRS with sleep, recent pains, smoking behaviour and whether there is any heart/cardiovascular condition showed the largest effect sizes. Various mechanisms can be involved in these behavioural patterns, such as hyper activity in the hypothalamic-pituitary-adrenal (HPA) axis⁷, and neurodevelopmental or parental impact on poorer health. Future research may explore the potentially bi-directional relationship between depression, related behavioural, neurobiological traits and poor physical health.”

Reviewer #3

Remarks to the Authors:

The authors have done an excellent job in addressing my reviewer comments. I have no further comments.

- In the interest of accountability and transparency during the review process, I choose to sign my review, Lea K. Davis.

We appreciate Reviewer #3 for her positive comments.

References

1. Franke, B. *et al.* Genetic influences on schizophrenia and subcortical brain volumes: large-scale proof of concept. *Nat. Neurosci.* **19**, (2016).
2. Length, T. *et al.* Accelerated Aging in Depression : From Physiological Aging to Brain Aging Chair : Brenda Penninx Co-Chair : Lianne Schmaal. *Biol. Psychiatry* **83**, S17 (2018).
3. Peyrot, W. J. *et al.* Does Childhood Trauma Moderate Polygenic Risk for Depression? A Meta-analysis of 5765 Subjects From the Psychiatric Genomics Consortium. *Biol. Psychiatry* **84**, 138–147 (2018).
4. Peyrot, W. J. *et al.* Effect of polygenic risk scores on depression in childhood trauma. *Br J Psychiatry* **205**, 113–119 (2014).
5. Howard, D. *et al.* Genome-wide meta-analysis of depression identifies 102 independent variants and highlights the importance of the prefrontal brain regions. *Nat. Neurosci.* **22**, 343–352 (2019).
6. Munafò, M. R., Tilling, K., Taylor, A. E., Evans, D. M. & Davey Smith, G. Collider scope: when selection bias can substantially influence observed associations. *Int. J. Epidemiol.* **47**, 226–235 (2018).
7. Pariante, C. M. & Lightman, S. L. The HPA axis in major depression: classical theories and new developments. *Trends Neurosci.* **31**, 464–468 (2008).

Reviewers' Comments:

Reviewer #4:

Remarks to the Author:

The authors carried out phenome-wide analyses of depression, including polygenic risk score and Mendelian randomization analyses, making use of the large-scale data collection of the UK Biobank study. Below are my comments in response to the author's responses to reviewer #1 and my remaining concerns.

1. Although the authors improved the discussion of the manuscript, I still think the discussion clearly misses a sufficient depth, which is important for understanding why the phenome-wide approach is most useful here. The discussion does not convince the reader why the phenome-wide approach should be considered over more thoughtful (traditional, and to some old-fashioned) epidemiological analyses, considering for each analysis which health-related confounders play a role and may explain the results (that I feel are missing in some of the associations that the authors find). Given the availability of a wide variety of variables that are available to us as researchers with an approved UK Biobank application, the paper feels like a 'we could so we should' approach without providing compelling arguments 'why we should'. Given the fact that the authors (like they mention multiple times in the discussion) are among the first to explore this approach, they should provide more arguments why a wideness of the approach should be preferred over depth.

Also, given the large sample sizes that we work with in UK Biobank, we should always be aware of the small individual effect sizes that can be picked up using well-powered samples, this should be mentioned in the discussion.

Although the discussion is already quite long as it is, it could be rewritten less lengthy and focusing on the more important discussion points that follow from their analyses (instead of mentioning what future studies should do).

2. A: I agree with the suggestion of reviewer #1 to add additional sample characteristics in supplementary table 17, since we know that the UK Biobank is not a representative reflection of the 'general population'. Given the wealth of phenotype data used in the paper, I think the characteristics that were chosen were quite general, and do not include any disease-related information, e.g. depression scores, depression diagnoses, medication use, etc. . In general, the supplementary table still provides us insufficient information about the overall health of the samples that were studied.

With regard to the reviewer #1's point 2 of multiple testing correction, the authors took the FDR approach which is more lenient than other correction methods. As a sensitivity analyses, other methods such as Bonferroni correction should be explored as well, to show the reader how many of the associations actually hold and are dependent on the choice of correction methods.

3. I agree with the authors that including scanner site as a covariate is a good approach here instead of running multi-level modeling, and the authors clearly show that this does not change the results/conclusions of the paper. Also, the main principal components already in the model probably already account for much of the effects that may be introduced by scanner. I have no further comments with regard to point 3 of reviewer #1.

4. Could the authors comment on how much of the association between depression PRS's (which are associated with many negative health outcomes, as the authors themselves show) and diffusion tensor imaging measures derived from DTI, is actually through these negative health effects and lifestyle factors (which are known to influence 'white matter health' as well and are not accounted for). Of course, molecular pathways like synaptic pruning processes the authors mention may be involved. However, it seems much more likely that the negative health effects that are 'captured' by these PRS's

play a role in less optimal DTI measure outcomes. Also, the authors should comment in the discussion section why the strongest association was found with MD rather than the other DTI measures, and what the authors say this would mean.

Point-to-point response

We would like to thank the editor and reviewers for their consideration of our manuscript. We have now addressed these comments in a revised version. A detailed point-by-point response is provided below. We have also taken care to consolidate responses over the rounds of reviews to provide a coherent manuscript.

Comments from the editor:

Many of the reviewers' comments are geared towards making the study (its aims and its conclusions) more accessible to readers but attention should also be paid to the question of the strength of the statistical analysis and potential confounding factors. Further, not just in the comments to the authors, but also in comments made to the editors, the reviewer has stressed the importance of justifying why you chose the approach (what does it add compared with other approaches?) and of including a balanced discussion of clinical utility (or a lack of it), especially in the light of small effect sizes, given the current interest in PRS (and likely hyping of anything PRS-related in the media). Please highlight all changes in the manuscript text file.

Response to the editor

We would like to thank the editor for considering our manuscript. A brief introduction to PheWAS approach and its strengths compared to other approaches has been added to the introduction on page 1.

“While genome-wide association studies (GWAS) seek to identify the many genetic associations of a single phenotype, a phenome-wide association study (PheWAS) reverses this approach to identify the multiple phenotypes associated with a single genetic risk score. This may be a stronger approach than studies that consider a single trait, based on prior theory, as PheWAS are less constrained by prior assumptions, particularly in situations where we currently have an incomplete understanding of disease mechanisms. Genotype based PheWAS approaches also have the considerable advantage that they are based on robust biological knowledge that is fixed from birth and therefore less susceptible to confounding and reverse causality.”

In addition, we have added a more balanced consideration of its strengths and weaknesses to the discussion, addressing reviewer #4, comment #1 in addition to the editor’s concerns.

We now avoid using any deterministic language to describe the genetic associations and the effect of genetic predisposition. In a previous response to reviewers, we highlighted the small effect sizes and the need to improve polygenic risk score accuracy through larger GWAS studies. This paragraph is highlighted on page 13, which reads:

“While PRS is a powerful means of identifying factors associated with genetic risk, it currently explains around 1.6% of phenotypic variance in depression. Future PRS scores, trained on more

precise GWAS summary statistics, are likely to be more strongly predictive and may have greater sensitivity to detect disease-relevant phenotypes. Further associations may be revealed as PheWAS studies increase in size, although this is counterbalanced by their small effect sizes and likely limited clinical utility for individual patients.”

Reviewer #4 (Remarks to the Author):

The authors carried out phenome-wide analyses of depression, including polygenic risk score and Mendelian randomization analyses, making use of the large-scale data collection of the UK Biobank study. Below are my comments in response to the author's responses to reviewer #1 and my remaining concerns.

1. Although the authors improved the discussion of the manuscript, I still think the discussion clearly misses a sufficient depth, which is important for understanding why the phenome-wide approach is most useful here. The discussion does not convince the reader why the phenome-wide approach should be considered over more thoughtful (traditional, and to some old-fashioned) epidemiological analyses, considering for each analysis which health-related confounders play a role and may explain the results (that I feel are missing in some of the associations that the authors find). Given the availability of a wide variety of variables that are available to us as researchers with an approved UK Biobank application, the paper feels like a 'we could so we should' approach without providing compelling arguments 'why we should'. Given the fact that the authors (like they mention multiple times in the discussion) are among the first to explore this approach, they should provide more arguments why a wideness of the approach should be preferred over depth.

We would like to thank reviewer #4 for their detailed comments. We have now added a summary paragraph discussing the advantages of the PheWAS approach (see the responses to the editor) in the introduction on page 1:

"While genome-wide association studies (GWAS) seek to identify the many genetic associations of a single phenotype, a phenome-wide association study (PheWAS) reverses this approach to identify the multiple phenotypes associated with a single genetic risk score. This may be a stronger approach than studies that consider a single trait, based on prior theory, as PheWAS are less constrained by prior assumptions, particularly in situations where we currently have an incomplete understanding of disease mechanisms. Genotype based PheWAS approaches also have the considerable advantage that they are based on robust biological knowledge that is fixed from birth and therefore less susceptible to confounding and reverse causality."

And in addition, we have added the following text to the discussion on page 12,

"While GWAS seeks to identify the genetic associations of a single phenotype, PheWAS reverses this approach to identify the multiple phenotypes associated with a single risk score. This is arguably a stronger approach than studies that consider a single trait, based on prior theory, as PheWAS is less constrained by prior assumptions based on an incomplete understanding of disease mechanisms. Genotype based PheWAS approaches also have the considerable advantage that they are based on robust biological knowledge that is fixed from birth and less susceptible to reverse causality."

Also, given the large sample sizes that we work with in UK Biobank, we should always be aware of the small individual effect sizes that can be picked up using well-powered samples, this should be mentioned in the discussion.

We agree and we have directly responded to this comment in our response above.

Although the discussion is already quite long as it is, it could be rewritten less lengthy and focusing on the more important discussion points that follow from their analyses (instead of mentioning what future studies should do).

Our discussion regarding the future research directions has been reduced in length. More in-depth interpretations of the present findings, as suggested by reviewer #4 in other comments, have been added and highlighted in the main text. In addition, we have re-read the entire discussion, combined duplicated information that was introduced through the multiple rounds of revision and improved readability throughout.

2. A: I agree with the suggestion of reviewer #1 to add additional sample characteristics in supplementary table 17, since we know that the UK Biobank is not a representative reflection of the 'general population'. Given the wealth of phenotype data used in the paper, I think the characteristics that were chosen were quite general, and do not include any disease-related information, e.g. depression scores, depression diagnoses, medication use, etc. . In general, the supplementary table still provides us insufficient information about the overall health of the samples that were studied.

We agree with the reviewer and we have now included disease-related phenotypes Supplementary Table 17 (see Supplementary Materials, page 43) as requested. The new characteristics include numbers of cases and controls for the three depression definitions, PHQ-4 for current depressive symptoms, self-reported use of anti-depressants. Educational attainment, which has been previously reported to be higher in UKB than in the general population, is currently detailed in the table.

With regard to the reviewer #1's point 2 of multiple testing correction, the authors took the FDR approach which is more lenient than other correction methods. As a sensitivity analyses, other methods such as Bonferroni correction should explored as well, to show the reader how many of the associations actually hold and are dependent on the choice of correction methods.

We agree that the addition of the Bonferroni threshold would provide further transparency and information on the statistical robustness of our findings. While the phenotypes are expected to be correlated with one another, justifying the use of FDR in accordance with previous publications, we have also added the p-value threshold for Bonferroni correction in the Supplementary Figures.

3. I agree with the authors that including scanner site as a covariate is a good approach here

instead of running multi-level modeling, and the authors clearly show that this does not change the results/conclusions of the paper. Also, the main principal components already in the model probably already account for much of the effects that may be introduced by scanner. I have no further comments with regard to point 3 of reviewer #1.

We agree with Reviewer #4's comments on the choice of covariates.

4. Could the authors comment on how much of the association between depression PRS's (which are associated with many negative health outcomes, as the authors themselves show) and diffusion tensor imaging measures derived from DTI, is actually through these negative health effects and lifestyle factors (which are known to influence 'white matter health' as well and are not accounted for). Of course, molecular pathways like synaptic pruning processes the authors mention may be involved. However, it seems much more likely that the negative health effects that are 'captured' by these PRS's play a role in less optimal DTI measure outcomes.

We agree that the association between depression PRS scores and measures of brain health may indeed be mediated through negative health effects and lifestyle factors. It is worth pointing out that this is not a case of confounding, but a question of how the effects identified in our analyses are mediated.

While we agree that the addition of multiple mediation analyses to the current manuscript would be revealing, whatever their findings, they would necessarily add further complexity and length. We hope the reviewer will not mind if we leave this to subsequent papers or investigators – as we think this necessitates very detailed consideration, perhaps focussed more narrowly than our strongly conservative phenome-wide approach.

Also, the authors should comment in the discussion section why the strongest association was found with MD rather than the other DTI measures, and what the authors say this would mean.

We agree that the biological significance of the larger PRS associations with MD should be further clarified, but unfortunately this is somewhat limited by our current cell and tissue-level understanding of diffusion metrics. We have however added the following text to the paper on page 10, adding further information in what is known on this subject and highlighting the importance of this issue:

“The strongest replicated white matter finding with PRS were found for the MD measures, consistent with previously reported depressive symptom associations¹. This may due to MD's greater sensitivity to ageing and related pathophysiological processes in this mid- to late-life UK Biobank sample². Alternatively, the associations with dispersion density suggest that reductions in MD may be partly due to reduced neurite density. This highlights the need for further investigation of these issues in tissue from large samples of depressed individuals.”

References

1. Shen, X. *et al.* White Matter Microstructure and Its Relation to Longitudinal Measures of Depressive Symptoms in Mid- and Late Life. *Biol. Psychiatry* **86**, 759–768 (2019).
2. Cox, S. R. *et al.* Ageing and brain white matter structure in 3,513 UK Biobank participants. *Nat. Commun.* **7**, 1–34 (2016).